# Adaptive Rollout Allocation for Online Reinforcement Learning with Verifiable Rewards

**Hieu Trung Nguyen**[C][*][†]**, Bao Nguyen**[C][†]**, Wenao Ma**[H][‡]**, Yuzhi Zhao**[H]**, Ruifeng She**[H]**, Viet Anh Nguyen**[C]

[C]The Chinese University of Hong Kong, [H]Huawei Hong Kong Research Center

{thnguyen,nbnguyen,nguyen}@se.cuhk.edu.hk
{ma.wenao,she.ruifeng}@huawei.com
yzzhao2-c@my.cityu.edu.hk
⭘ **Source Code:** https://github.com/HieuNT91/VIP

## Abstract

Sampling efficiency is a key bottleneck in reinforcement learning with verifiable rewards. Existing group-based policy optimization methods, such as GRPO, allocate a fixed number of rollouts for all training prompts. This uniform allocation implicitly treats all prompts as equally informative, and could lead to inefficient computational budget usage and impede training progress. We introduce VIP, a Variance-Informed Predictive allocation strategy that allocates a given rollout budget to the prompts in the incumbent batch to minimize the expected gradient variance of the policy update. At each iteration, VIP uses a lightweight Gaussian process model to predict per-prompt success probabilities based on recent rollouts. These probability predictions are translated into variance estimates, which are then fed into a convex optimization problem to determine the optimal rollout allocations under a hard compute budget constraint. Empirical results show that VIP consistently improves sampling efficiency and achieves higher performance than uniform or heuristic allocation strategies in multiple benchmarks.

## 1 Introduction

The advent of Language Models (LMs) has marked a new era in artificial intelligence, transforming models from static knowledge bases into dynamic, intelligent agents capable of complex reasoning, creative generation, and intricate problem-solving. This evolution has been largely driven by the advancement of post-training methods (Tong et al., 2024; Rafailov et al., 2023; Shao et al., 2024) and test-time scaling methods (Snell et al., 2024; Feng et al., 2023; Nguyen et al., 2025a;b), which are crucial for aligning a base LM's vast, general knowledge with specific human intentions, ethical guidelines, and desired task-oriented behaviors. These refinement processes bridge the gap between a model that understands language and one that can reliably and safely act as a collaborator, simulator, or autonomous agent. As LMs become more integrated into real-world applications, the effectiveness and efficiency of these post-training stages have become a primary focus of research and development.

While test-time scaling methods are training-free and effective in many scenarios, post-training generally yields higher performance gains at a lower inference budget. Two prominent paradigms are used for post-training LMs: Supervised Fine-Tuning (SFT) and Reinforcement Learning (RL). SFT is a computationally efficient approach that trains the model on a curated dataset of input-output pairs, effectively equipping it with domain-specific knowledge. However, its effectiveness is often hampered by the difficulty of curating high-quality and comprehensive datasets. In contrast, RL-based methods, which treat the model as an agent optimizing for a reward signal derived from human preferences (Reinforcement Learning from Human Feedback, RLHF) or external verification

---

[*]Work done during an internship at Huawei Hong Kong Research Center
[†]Equal contribution
[‡]Corresponding author

mechanisms (Reinforcement Learning from Verifiable Rewards, RLVR), can explore a broader action space and achieve superior performance on complex, open-ended tasks. RLVR, in particular, has gained wide popularity thanks to its low reliance on human input. However, the performance of RL methods comes at a significant computational cost. Algorithms like Proximal Policy Optimization (PPO, Schulman et al. (2015)) require the simultaneous training of a separate value model to estimate the advantage of actions, which can be prohibitively memory-intensive in resource-constrained situations. To address this, a family of "critic-free" or group-based RL methods has emerged, estimating the advantage as a relative measure within a group of rollouts, essentially trading speed for memory efficiency. Such methods include GRPO, (Shao et al., 2024), Dr. GRPO (Liu et al., 2025), RLOO (Ahmadian et al., 2024), and other variants. A direct drawback of these methods is the additional computation time required to generate multiple rollouts for a training example. The number of generations often needs to be large enough (e.g., 16) to obtain a stable training process, which exacerbates the generation overhead and eventually leads to a memory-bound system. Such high computational demands consequently highlight the need for greater sampling efficiency in the RL training process.

The high computational demands of RL algorithms have led to a body of work dedicated to optimizing this performance-cost trade-off. One line of research involves hybrid approaches that combine an initial SFT phase with a subsequent RL refinement stage (Chen et al., 2025a) or merge the two stages into a single method (Fu et al., 2025). These methods demonstrated improvement in smoothening or even accelerating the training process, but often fail to dynamically adapt to the training process based on the model's evolving capabilities for solving the training problems. The observation that simply filtering out problems where the accuracy is close to either 0 or 1 can boost performance (Yu et al., 2025) hints at the potential for more adaptive control. To the best of our knowledge, there is a lack of crucial metrics for such capabilities and, consequently, a lack of an adaptive control mechanism for the GRPO process.

To address such challenges, we introduce the Variance-Informed Predictive allocation strategy (VIP), a principled training framework for efficient and adaptive rollout allocation in group-based policy optimization. We first provide a theoretical analysis of the gradient variance for three prominent RL algorithms, establishing the relationship between the gradient variance contributed by a prompt and the probability that it is solved by the current model. Building on this insight, at the beginning of each training iteration, VIP predicts the expected gradient variance of each prompt in a mini-batch. Based on these predictions, VIP then solves a convex optimization problem followed by integer rounding heuristic to allocate rollouts across prompts to minimize the total expected gradient variance under the given computational budget. The core contributions of this paper are the following:

- **Gradient variance analysis.** We provide a rigorous analysis of the effect of gradient variance on the RL training process. We derive the connection between gradient variance and success probability for prevailing group-based RL methods, including Dr. GRPO and RLOO. This establishes the theoretical foundation for adaptively controlling budget allocation during training.

- **Variance prediction.** VIP employs a Gaussian process (GP) over prompt embeddings to model the probability of success for each prompt in any training step. This enables recursive Bayesian updates that leverage both past rollout outcomes and the similarity structure of the prompts. The GP prediction allows the framework to estimate the gradient variance for each prompt in the minibatch at every training step.

- **Variance-minimizing rollout allocation.** Given the predicted variances, VIP formulates a convex optimization problem to determine the optimal allocation of rollouts on prompts. This allocation minimizes the gradient variance subject to a rollout budget constraint. We derive an efficient algorithm that provides the exact solution to the continuous relaxation and further develop a greedy incentive-based rounding heuristic to produce a feasible integer solution. This process enables fast resource allocation under computational budget constraints.

Our paper unfolds as follows. Section 2 discusses related work on adaptive strategies over group-based RL frameworks for RLVR, while Section 3 lays the technical background on RLVR. Section 4 presents our analysis of the per-prompt gradient variance. Section 5 delineates our VIP framework for variance prediction via Gaussian process, and the optimization problem for rollout allocation. Section 6 presents empirical results on mathematical reasoning and information retrieval datasets.

## 2 RELATED WORK

Recent advancements in group-based RL for LMs have focused on adaptive strategies to enhance both training efficiency and performance. These methods move beyond static training pipelines by dynamically selecting and managing data used for policy updates. Initial research empirically demonstrated that filtering out "non-informative" problems to which the model's accuracy is either 0 or 1 can improve training efficiency, as these problems lead to rollout batches with zero variance and consequently make no contribution to the gradient signal (Yu et al., 2025). Along similar lines, (Lin et al., 2025) filtered out rollouts with low absolute advantage, while (Xu et al., 2025) proposed to retain examples with the highest variance. Lacking an a priori measure of training prompt informativeness, these methods require the pre-sampling of an large batch of rollouts before each update iteration, potentially negating efficiency gains from the significant sampling overhead.

Several works adopted a pre-rollout estimation of the prompt informativeness. (Zheng et al., 2025) proposed skipping each problem with a probability determined by the number of recent consecutive rollouts where the prompt is non-informative. (Zhang et al., 2025) proposed a signal-to-noise ratio indicating a problem's contribution to the gradient, and showed that training can be accelerated by generating a small trial batch of rollouts to identify and filter out non-informative problems, and then continue sampling on intermediate-difficulty ones. (Sun et al., 2025) designed a difficulty-targeted online selection algorithm with attention-based difficulty prediction and rollout replay. (Liao et al., 2025) ranked problems by difficulty, indicated by the average rewards from past rollouts, and allocated resources to sample larger batches for difficult prompts. (Yang et al., 2025) took a similar approach as (Zhang et al., 2025) to estimate problem difficulty, and assigned a larger budget and higher gradient weights to difficult prompts. (Kong et al., 2025) estimates prompt difficulty by aggregating historical performance discrepancies of the problems, then adaptively selects the set of problems whose difficulty is in alignment with the current competence of the model. Other works also considered the entropy-reward trade-off to mitigate the risk of premature convergence. (Liao et al., 2025) combined difficulty-aware reallocation with entropy-stabilizing temperature scheduling to effectively balance efficiency and exploration. Another approach steers exploration by maximizing information gain throughout the training process (Lee et al., 2024).

## 3 BACKGROUND ON RLVR

We study the prominent family of methods for RL training that employs group-based advantage estimation to stabilize learning and better utilize reward signals. This family includes Group Relative Policy Optimization (GRPO)(Shao et al., 2024) and its variants, such as Dr. GRPO (Liu et al., 2025) and RLOO (Ahmadian et al., 2024). Given a dataset of $Q$ prompts $\mathcal{Q} = \{q_1, q_2, \ldots, q_Q\}$, the general objective function for group-based policy optimization methods is

$$J(\theta) = \mathbb{E}_{q \sim \mathcal{Q}} \mathbb{E}_{\pi_{\text{old}}^{\otimes n}} \left[ \frac{1}{n} \sum_{j=1}^{n} \frac{1}{|\tilde{o}_j|} \sum_{\tau=1}^{|\tilde{o}_j|} \left( A'_{j,\tau} - \beta \, D_{\text{KL}} \big( \pi_\theta(\cdot \mid q, \tilde{o}_{j,<\tau}) \, \big\| \, \pi_{\text{ref}}(\cdot \mid q, \tilde{o}_{j,<\tau}) \big) \right) \right], \quad (1)$$

where $n$ is the number of rollouts for each prompt, $\tilde{o}_j$ is the $j$-th rollout from policy $\pi_{\text{old}}$, $|\tilde{o}_j|$ is the number of tokens in $\tilde{o}_j$, and $\pi_\theta$ is the current policy with its learnable parameters $\theta$. Here, $D_{\text{KL}} \big( \pi_\theta(\cdot \mid q, \tilde{o}_{j,<\tau}) \, \big\| \, \pi_{\text{ref}}(\cdot \mid q, \tilde{o}_{j,<\tau}) \big)$ is the Kullback-Leibler divergence between the policy $\pi_\theta$ and a reference policy $\pi_{\text{ref}}$, both of which are conditioned on the prompt $q$ and the tokens generated thus far $\tilde{o}_{j,<\tau}$. We use a tilde ($\sim$) to emphasize the stochastic nature of the quantities.

For prompt $q$, the term $A'_{j,\tau}$ is the clipped surrogate at token $\tau$ for output $\tilde{o}_j$:

$$A'_{j,\tau} = \min \left( r_{j,\tau} \tilde{A}_{j,\tau}, \; \text{clip}(r_{j,\tau}, 1 - \epsilon, 1 + \epsilon) \, \tilde{A}_{j,\tau} \right), \quad \text{with} \quad r_{j,\tau}(\theta) = \frac{\pi_\theta(\tilde{o}_{j,\tau} \mid q, \tilde{o}_{j,<\tau})}{\pi_{\text{old}}(\tilde{o}_{j,\tau} \mid q, \tilde{o}_{j,<\tau})}. \quad (2)$$

Notice that we momentarily omit the index $q$ to avoid clutter. Here, $r_{j,\tau}$ is the relative ratio between current and data-generating policies; while $\tilde{A}_{j,\tau}$ is the advantage estimator for token $\tau$ in output $\tilde{o}_j$. Let $R(\tilde{o}_j)$ be the reward for output $\tilde{o}_j$ on prompt $q$, taking values of $-1$ (incorrect) or $1$ (correct). Based on the whole set of $n$ rollouts $\{\tilde{o}_j\}$, there are two popular advantage estimators:

- In Dr. GRPO (Liu et al., 2025), $\tilde{A}_j = \tilde{A}_{j,\tau} = R(\tilde{o}_j) - \frac{1}{n} \sum_k R(\tilde{o}_k)$, which uses all rollouts to compute the mean.

- In RLOO (Ahmadian et al., 2024), $\tilde{A}_j = \tilde{A}_{j,\tau} = R(\tilde{o}_j) - \frac{1}{n-1}\sum_{k \neq j} R(\tilde{o}_k)$, which excludes the $j$-th output when computing the mean.

Both advantage estimators are token-independent: all tokens in the $j$-th rollout admit the same advantage $\tilde{A}_j$. In this paper, we will focus on a particular training regime under the next assumption.

**Assumption 3.1.** *We set the KL regularization term to zero, i.e., $\beta = 0$.*

Assumption 3.1 is both practical and non-restrictive for the following reasons. In RLHF, reward models are only reliable near the distribution of the reference policy (Gao et al., 2023; Chen et al., 2024; Huang et al., 2024; Ramé et al., 2024), which makes KL regularization essential to prevent reward hacking (Laidlaw et al., 2024; Song et al., 2024; Skalse et al., 2022). In contrast, our work focuses on the RLVR setting, where rewards are computed by rule-based verifiers. These verifiable rewards eliminate concerns about reward miscalibration. Indeed, recent studies in RLVR have successfully removed the KL term while still achieving state-of-the-art performance (Yu et al., 2025; Liu et al., 2025).

Assumption 3.1 simplifies the training objective by eliminating the KL term. The objective function reduces to:

$$J(\theta) = \mathbb{E}_{q \in \mathcal{Q}} \mathbb{E}_{\pi_{\text{old}}^{\otimes n}} \left[ \frac{1}{n} \sum_{j=1}^{n} \frac{1}{|\tilde{o}_j|} \sum_{\tau=1}^{|\tilde{o}_j|} \min\left( r_{j,\tau} \tilde{A}_j, \ \text{clip}(r_{j,\tau}, 1 - \epsilon, 1 + \epsilon)\, \tilde{A}_j. \right) \right]$$

Whenever $r_{j,\tau}$ falls outside the clipping region in a direction that worsens the surrogate, $\min\left( r_{j,\tau} \tilde{A}_j, \ \text{clip}(r_{j,\tau}, 1 - \epsilon, 1 + \epsilon)\, \tilde{A}_j \right)$ collapses to the clipped term and contributes no gradient. To isolate precisely when the gradient is active, we introduce the indicator

$$\mathbb{I}_{j,\tau}^{\text{unc}} = 1 - \left( \mathbf{1}\{r_{j,\tau} > 1 + \varepsilon\} \, \mathbf{1}\{\tilde{A}_j \geq 0\} + \mathbf{1}\{r_{j,\tau} < 1 - \varepsilon\} \, \mathbf{1}\{\tilde{A}_j < 0\} \right).$$

Using this indicator, we define $\bar{r}_j = \frac{1}{|\tilde{o}_j|} \sum_{\tau=1}^{|\tilde{o}_j|} \mathbb{I}_{j,\tau}^{\text{unc}} r_{j,\tau}$ and rewrite the objective as:

$$J(\theta) = \mathbb{E}_{q \in \mathcal{Q}} \mathbb{E}_{\pi_{\text{old}}^{\otimes n}} \left[ \frac{1}{n} \sum_{j=1}^{n} \tilde{A}_j \bar{r}_j \right] = \mathbb{E}_{q \in \mathcal{Q}} \mathbb{E}_{\pi_{\text{old}}^{\otimes n}} \left[ \frac{1}{n} \sum_{j=1}^{n} A_j(\{\tilde{o}_k\}) \bar{r}_j \right]. \tag{3}$$

The next section studies the gradient variance of $J$.

## 4 ANALYSIS FOR GRADIENT VARIANCE

In this section, we analyze the variance of the gradient estimator arising from sampling rollouts in RLVR algorithms such as GRPO and RLOO. Each update step combines two types of data: previously collected rollouts from past batches (off-policy) and newly generated rollouts for the current batch of prompts $\mathcal{B}$ (on-policy). However, as discussed in the subsequent Section 5, our research objective is to determine how many rollouts should be allocated to the prompts in $\mathcal{B}$. This allocation decision influences only the on-policy component of the gradient. The off-policy contributions, while present in the overall update, are determined entirely by past data and therefore remain unaffected by the number of new rollouts we choose to sample. Consequently, to understand how allocation impacts gradient variance, it is both natural and sufficient to isolate the variance arising from the on-policy rollouts generated by the current policy $\pi_\theta$. For these on-policy samples, the behavior and target policies coincide $\pi_{\text{old}} = \pi_\theta$, hence $\bar{r}_j = 1$.

The gradient contribution of these on-policy rollouts is:

$$\nabla_\theta J^{\text{on}}(\theta) = \mathbb{E}_{q \in \mathcal{B}} \mathbb{E}_{\pi_\theta^{\otimes n}} \left[ \frac{1}{n} \sum_{j=1}^{n} A_j(\{\tilde{o}_k\}) \underbrace{\frac{1}{|\tilde{o}_j|} \sum_{\tau=1}^{|\tilde{o}_j|} \nabla_\theta \log \pi_\theta(\tilde{o}_{j,\tau} | q, \tilde{o}_{j,<\tau})}_{\triangleq H(\tilde{o}_j)} \right]. \tag{4}$$

Above, $H(\tilde{o}_j)$ is the average log-likelihood gradient, where the average is taken over all tokens in the $j$-th rollout. Consider a specific prompt $q$ and the $n$ rollouts $\{\tilde{o}_k\}$, the contribution of $q$ to the

calculation of the sample gradient is $\frac{1}{n}\sum_{j=1}^{n} A_j(\{\tilde{o}_k\})H(\tilde{o}_j)$, and we omit the subscript $q$ to avoid clutter. This quantity is a random vector because $H(\tilde{o}_j)$ is a random vector. To make the variance analysis mathematically tractable while accurately proxying the scale of the parameter updates, we will study the $L_2$ norm of this gradient gradient contribution. To this end, we are interested in the (univariate) random variable:

$$\tilde{G} \triangleq \frac{1}{n}\sum_{j=1}^{n} A_j(\{\tilde{o}_k\})\|H(\tilde{o}_j)\|_2.$$

To simplify the notation, we define the random variables $\tilde{R}_j = R(\tilde{o}_j)$ and $\tilde{Z}_j = \|H(\tilde{o}_j)\|_2$ for all $j$. We first consider the Dr. GRPO case. The gradient $\tilde{G}$ has the specific form:

$$\tilde{G} \triangleq \frac{1}{n}\sum_{j=1}^{n} \big(\tilde{R}_j - \frac{1}{n}\sum_{k=1}^{n}\tilde{R}_k\big)\tilde{Z}_j.$$

To have a rigorous analysis of the variance of $\tilde{G}$, we make the following assumption explicitly.

**Assumption 4.1.** *(i) $\tilde{R}_1, \ldots, \tilde{R}_n$ are independent and identically distributed (i.i.d.) copies of $\tilde{R}$, and $\tilde{Z}_1, \ldots, \tilde{Z}_n$ are also i.i.d. copies of $\tilde{Z}$,*

*(ii) $\{\tilde{Z}_j\}$ and $\{\tilde{R}_j\}$ are uncorrelated up to second-order: $\mathrm{Cov}(\tilde{R}_k, \tilde{Z}_j) = 0$ for any $(k, j)$ and $\mathrm{Cov}(\tilde{R}_k\tilde{R}_{k'}, \tilde{Z}_j\tilde{Z}_{j'}) = 0$ for any $(k, j, k', j')$*

Assumption 4.1 is realistic in common settings: given the same prompt and fixed sampling procedure, different rollouts are independent samples from the same distribution $\pi_\theta$. Thus, it is reasonable to assume that $\{\tilde{o}_j\}$ are i.i.d., which induces Assumption 4.1(i). Assumption 4.1(ii) needs further justification because for the $j$-th rollout, the reward $\tilde{R}_j$ may be correlated with the average gradient $\tilde{Z}_j$. However, we have strong reasons to believe that it holds because $\tilde{R}$ is a Bernoulli distribution, and $\tilde{Z}_j$ aggregates variations both over the number of tokens (from the definition of $H(\tilde{o}_j)$) and across the high-dimensional parameter space (from $\|H(\tilde{o}_j)\|_2$). Empirically, we include in Appendix B a statistical test to support our assumption.

The next result establishes the variance for the Dr. GRPO case.

**Proposition 4.2** (Dr. GRPO gradient variance). *Consider a prompt with binary reward $R(\tilde{o}_j) \in \{1, -1\}$ with $\mathbb{P}(R(\tilde{o}_j) = 1) = p$. If Assumption 4.1 holds and the variance of the projected gradient $\tilde{Z}$ is $\sigma_Z^2$, the variance of the per-prompt projected Dr. GRPO gradient estimator with $n$ rollouts is*

$$\mathrm{Var}(\tilde{G}) = \frac{(n-1)}{n^2} 4\sigma_Z^2 p(1-p).$$

We now switch gears to the RLOO case. The RLOO gradient $\tilde{G}$ has the specific form:

$$\tilde{G} \triangleq \frac{1}{n}\sum_{j=1}^{n} \big(\tilde{R}_j - \frac{1}{n}\sum_{\substack{k=1\\k\neq j}}^{n}\tilde{R}_k\big)\tilde{Z}_j.$$

**Proposition 4.3** (RLOO gradient variance). *Consider a prompt with binary reward $R(\tilde{o}_j) \in \{1, -1\}$ with $\mathbb{P}(R(\tilde{o}_j) = 1) = p$. If Assumption 4.1 holds and the variance of the projected gradient $\tilde{Z}$ is $\sigma_Z^2$, the variance of the per-prompt projected RLOO gradient estimator with $n$ rollouts is*

$$\mathrm{Var}(\tilde{G}) = \frac{1}{n-1} 4\sigma_Z^2 p(1-p).$$

## 5 PREDICTIVE ROLLOUT ALLOCATION STRATEGY

Consider a specific iteration $t$ when we have drawn a mini-batch $\mathcal{B}_t$ from the set of training prompts. Propositions 4.2-4.3 indicate that the gradient variance for a prompt depends on the number of rollouts. Consequently, a uniform allocation of $n$ rollouts for every prompt could be inefficient. Section 4 reveals that the per-prompt gradient variance depends on the success probability $p_q$, which is the probability that model $\theta_t$ answers prompt $q$ correctly. However, this probability is not observable ex ante without performing rollouts, and we need to resort to an estimate $\hat{p}_q$ to make allocation

decisions. Building an estimator for the success probability is challenging for at least two reasons: First, the success probability is, unfortunately, not static: it depends on the model weights, which evolve after every training iteration. As the weights are updated, the distribution over the outputs changes, and consequently, the success probability drifts. Second, prompt embeddings may not contain sufficiently informative signal for parametric classifiers.

To improve rollout efficiency, we propose VIP, the Variance-Informed Predictive allocation strategy, which minimizes the minibatch gradient variance by allocating a different number of rollouts $\{n_q\}$ for each prompt in the minibatch. The complete workflow of VIP, from prediction to optimized rollout allocation, is illustrated in Figure 1. VIP comprises two main components: (i) a nonparametric Gaussian process model for predicting the success probability of the current model on each training prompt (green boxes), and (ii) an optimization model for making optimal allocation decisions given the computing budget (blue box). At the beginning of iteration $t$, we first predict the success probability for all $q \in \mathcal{B}_t$, as described in Section 5.1. The probability estimates serve as input to our optimization framework for allocating rollouts across prompts in Section 5.2.

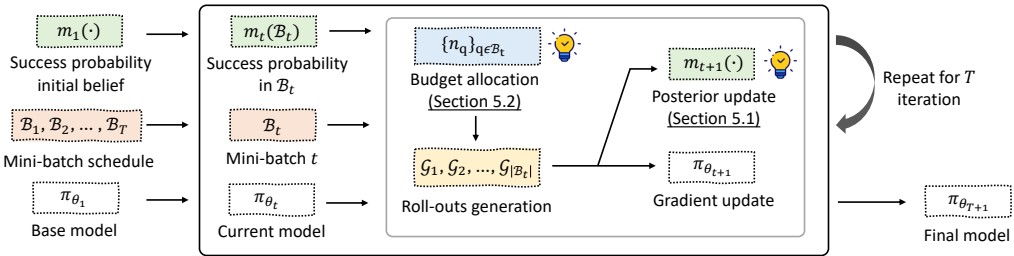

Figure 1: The process starts with an initial belief over prompt success probabilities. At each step $t$, a mini-batch $\mathcal{B}_t$ is selected, and the belief function $m_t(\cdot)$ predicts the success probabilities of the prompts in $\mathcal{B}_t$. A budget allocation module assigns rollout budgets $\{n_q\}$, rollouts are generated, and the resulting data updates the model and beliefs. Repeated for $T$ steps, this yields a fine-tuned model $\pi_{\theta_{T+1}}$ with improved performance and efficient rollout usage.

## 5.1 GP Prior and Recursive Posterior Update

Given the training prompts $\mathcal{Q} = \{q_1, q_2, \ldots, q_Q\}$, we denote their embeddings as $\mathcal{D} = \{x_q\}_{q=1}^Q$. At every iteration $t$, the success probability of the current model on prompt $q$ is modeled using a sigmoid link function on the latent value $g_t(x_q)$:

$$p_{q,t} = \mathrm{sigmoid}(g_t(x_q)) = 1/[1 + \exp(-g_t(x_q))] \in (0,1),$$

where $g_t : \mathbb{R}^d \to \mathbb{R}$ is a latent function over prompt embeddings. Because $g_t$ is real-valued, it is simpler to place a Gaussian process on $g_t$. Toward this goal, we set $g_t \sim \mathrm{GP}(m_t(\cdot), \mathcal{K}(\cdot, \cdot))$, where $m_t$ is the mean function and $\mathcal{K}$ is a radial basis function (RBF) kernel with bandwidth $h > 0$: $\mathcal{K}(x, x') = \exp(-\|x - x'\|_2^2/(2h^2))$.

**Initialization.** At $t = 1$, we use a zero-mean prior $g_1 \sim \mathrm{GP}(0, \mathcal{K})$. Over the dataset $\mathcal{D} = \{x_q\}_{q=1}^Q$, this gives $g_1(\mathcal{D}) \sim \mathcal{N}(0, \Sigma)$ where $\Sigma = \mathcal{K}(\mathcal{D}, \mathcal{D})$ is the kernel matrix over all prompts.

**Prediction.** At iteration $t$, the prior $m_t$ is used to predict $\hat{p}_{q,t} = \mathrm{sigmoid}(m_t(x_q))$ for $q \in \mathcal{B}_t$. These predicted values will be sent to the optimization problem to compute the rollout allocation.

**Sequential updates.** At iteration $t \geq 1$, the latent values for all prompts are captured by a random vector $[g_t(x_1), \ldots, g_t(x_Q)]^\top \sim \mathcal{N}(m_t, \Sigma)$. For $q \in \mathcal{B}_t$, we observe $n_q$ rollouts and rewards $\tilde{R}_{q,j} \in \{-1, 1\}$ for $j = 1, \ldots, n_q$. Then, we compute the clipped average:

$$\bar{R}_q = \frac{1}{n_q} \sum_{j=1}^{n_q} \tilde{R}_{q,j} \in [-1, 1], \quad \hat{p}_{q,t} = \mathrm{clip}\left(\frac{\bar{R}_q + 1}{2}, \epsilon, 1 - \epsilon\right) \in (\epsilon, 1 - \epsilon),$$

which induces an observation for the latent value $\hat{g}_{q,t} = \mathrm{sigmoid}^{-1}(\hat{p}_{q,t}) \in \mathbb{R}$ for $q \in \mathcal{B}_t$. Clipping is necessary here to avoid the situation when $\bar{R}_{q,t} = -1$ or $1$, which leads to infinite values for the latent variable. The collection of these latent values is denoted $\hat{g}_{\mathcal{B}_t}$. Let $\mathcal{B}_t^c$ be the complement set

containing all prompts that are *not* in the mini-batch $\mathcal{B}_t$. We can partition the mean vector and the covariance matrix into the block form:

$$m_t = \begin{bmatrix} m_{t,\mathcal{B}_t} \\ m_{t,\mathcal{B}_t^c} \end{bmatrix}, \qquad \Sigma = \begin{bmatrix} \Sigma_{\mathcal{B}_t\mathcal{B}_t} & \Sigma_{\mathcal{B}_t\mathcal{B}_t^c} \\ \Sigma_{\mathcal{B}_t^c\mathcal{B}_t} & \Sigma_{\mathcal{B}_t^c\mathcal{B}_t^c} \end{bmatrix},$$

the posterior over unqueried prompts at iteration $t$ is $g_{t,\mathcal{B}_t^c} \mid \hat{g}_{\mathcal{B}_t} \sim \mathcal{N}(m_t^\star, \Sigma^\star)$ with

$$m_{t,\mathcal{B}_t^c}^\star = m_{t,\mathcal{B}_t^c} + \Sigma_{\mathcal{B}_t^c\mathcal{B}_t} \Sigma_{\mathcal{B}_t\mathcal{B}_t}^{-1} (\hat{g}_{\mathcal{B}_t} - m_{t,\mathcal{B}_t}), \quad \Sigma^\star = \Sigma_{\mathcal{B}_t^c\mathcal{B}_t^c} - \Sigma_{\mathcal{B}_t^c\mathcal{B}_t} \Sigma_{\mathcal{B}_t\mathcal{B}_t}^{-1} \Sigma_{\mathcal{B}_t\mathcal{B}_t^c}.$$

**Next-step prior.** The prior mean at iterate $t$ is updated into the posterior mean as follows: we set $m_{t+1}(x_q) = \hat{g}_{q,t}$ if $q \in \mathcal{B}_t$ and set $m_{t+1}(x_q) = m_{t,\mathcal{B}_t^c}^\star(x_q)$ if $q \in \mathcal{B}_t^c$. This posterior mean $m_{t+1}$ will serve as the prior in iteration $t + 1$.

## 5.2 Adaptive Budget Allocation for Gradient Variance Minimization

Consider a mini-batch $\mathcal{B}_t$ of $B$ prompts, and every prompt $q \in \mathcal{B}_t$ has a *predicted* success probability $\hat{p}_q$ under the GP model from Section 5.1. We now focus on deciding the number of rollouts $\{n_q\}$ assigned to each prompt in this batch $\mathcal{B}_t$ under a hard constraint of a total budget of $C$ rollouts. Additionally, we impose that the number of rollouts should be between a lower bound $L$ and an upper bound $U$: if the number of rollouts is too small, it is unlikely to obtain discriminative reward signals, while a large number of rollouts may lead to overfitting. For a meaningful setting, we require $L \geq 3$, and assume that $BL \leq C \leq BU$.

Our objective is to minimize the sum of the gradient variance induced by prompts in the mini-batch $\mathcal{B}_t$. This problem can be formulated as an integer optimization problem:

$$\min \left\{ \sum\nolimits_{q\in\mathcal{B}_t} \text{Var}(\tilde{G}_q) \ : \ \sum\nolimits_{q\in\mathcal{B}_t} n_q = C, \quad n_q \in \{L, L+1, \ldots, U\} \quad \forall q \in \mathcal{B}_t \right\}. \tag{5}$$

The variance $\text{Var}(\tilde{G}_q)$ is likely a nonlinear function of the rollout allocation $n_q$, and (5) becomes a nonlinear integer optimization problem. To tackle this problem, we will first solve its continuous relaxation and then apply a heuristic rounding algorithm.

**Relaxed Solution for Budget Allocation.** Proposition 4.2 provides the per-prompt gradient variance $\text{Var}(\tilde{G}_q)$ for Dr. GRPO. By defining $a_q \triangleq 4\sigma_{Z_q}^2 \hat{p}_q(1 - \hat{p}_q)$ for every $q \in \mathcal{B}_t$, the continuous relaxation of problem (5) for Dr. GRPO is

$$\min \left\{ \sum\nolimits_{q\in\mathcal{B}_t} a_q \frac{n_q - 1}{n_q^2} \ : \ \sum\nolimits_{q\in\mathcal{B}_t} n_q = C, \quad L \leq n_q \leq U, \quad n_q \in \mathbb{R} \ \forall q \in \mathcal{B}_t \right\}. \tag{6}$$

The next theorem asserts a computationally efficient method to solve (6).

**Theorem 5.1** (Continuous allocation, Dr. GRPO). *For each $q \in \mathcal{B}_t$, let $n_q^\star$ be the function:*

$$n_q^\star(\lambda) = \begin{cases} U & \text{if } \lambda \leq a_q \frac{U-2}{U^3}, \\ \text{the unique solution to } \lambda = a_q \frac{n_q - 2}{n_q^3} & \text{if } a_q \frac{U-2}{U^3} < \lambda < a_q \frac{L-2}{L^3}, \\ L & \text{if } \lambda \geq a_q \frac{L-2}{L^3}. \end{cases} \tag{7}$$

*If $BL \leq C \leq BU$, the algebraic equation $\sum_q n_q^\star(\lambda^\star) = C$ has a unique solution $\lambda^\star$. Then the unique minimizer of (6) is given by $n_q^\star = n_q^\star(\lambda^\star)$ for all $q$. Moreover, $\lambda^\star$ can be found by bisection.*

Similarly, for the RLOO case, we can leverage Proposition 4.3 to define the same parameters $a_q \triangleq 4\sigma_{Z_q}^2 \hat{p}_q(1 - \hat{p}_q)$ for $q \in \mathcal{B}_t$. The continuous relaxation of (5) for RLOO is

$$\min \left\{ \sum\nolimits_{q\in\mathcal{B}_t} a_q \frac{1}{n_q - 1} \ : \ \sum\nolimits_{q\in\mathcal{B}_t} n_q = C, \quad L \leq n_q \leq U, \quad n_q \in \mathbb{R} \ \forall q \in \mathcal{B}_t \right\}. \tag{8}$$

We can solve (8) efficiently thanks to the next result.

**Theorem 5.2** (Continuous allocation, RLOO). *For each $q \in \mathcal{B}_t$, let $n_q^\star$ be the function:*

$$n_q^\star(\lambda) = \begin{cases} U & \text{if } \lambda \leq a_q/(U-1)^2, \\ 1 + \sqrt{a_q/\lambda} & \text{if } a_q/(U-1)^2 < \lambda < a_q/(L-1)^2, \\ L & \text{if } \lambda \geq a_q/(L-1)^2 \end{cases} \quad (9)$$

*If $BL \leq C \leq BU$, the algebraic equation $\sum_q n_q^\star(\lambda^\star) = C$ has a unique solution $\lambda^\star$. Then the unique minimizer of (8) is given by $n_q^\star = n_q^\star(\lambda^\star)$ for all q. Moreover, $\lambda^\star$ can be found by bisection.*

**Heuristic Rounding.** First, $n_q^\star$ is rounded down to $\lfloor n_q^\star \rfloor$. Let $f_q(n)$ denote the per-prompt objective: $f_q(n) = a_q \frac{n-1}{n^2}$ for Dr. GRPO and $f_q(n) = a_q \frac{1}{n-1}$ for RLOO. The remaining budget is then distributed iteratively to prompts with the largest decrease in $f_q$ if an additional rollout is allocated. This process continues until the total budget $C$ is exhausted, while ensuring that the per-prompt bounds, $L \leq \hat{n}_q \leq U$, are satisfied. The procedure is detailed in Appendix D.

## 6 NUMERICAL EXPERIMENTS

We now showcase the empirical benefit of our VIP framework on the mathematical reasoning task and the tool-augmented reasoning task. The common setup for both tasks is as follows: we encode each prompt $q$ into a 384-dimensional vector $x_q$ using `all-MiniLM-L6-v2`. Before training, we pre-compute and cache the pairwise Euclidean distances between all prompt pairs, making the runtime overhead negligible. We use a median heuristic to set the bandwidth $h$ of the kernel $\mathcal{K}$. We also assume that $\tilde{Z}_q$ has the same variances over all $q$ when we compute the allocation; we support this assumption with a statistical test presented in Appendix B.3. We train all models for two epochs under two total rollout budgets, $8 \times Q$ and $16 \times Q$, where $Q$ is the dataset size. We integrate VIP on top of two group-relative policy optimization baselines, RLOO and Dr. GRPO, implemented using VERL (Sheng et al., 2024) for math experiments and following (Chen et al., 2025b; Jin et al., 2025) for tool-augmented tasks. We closely follow the default hyperparameter settings used in RL training.

**Mathematical Reasoning Task.** For mathematical reasoning, we train on DAPO-MATH-17k (Yu et al., 2025) and evaluate on AIME2024 and AIME2025. We evaluate VIP by comparing Dr. GRPO and RLOO, each with and without our adaptive rollout allocation method. We experiment with three backbone LMs (`Qwen2.5-Math-1.5B`, `Qwen2.5-Math-7B`, and `Llama-3.2-3B-Instruct`), on AIME24/25, under two rollout-budget settings $C \in \{8 \times Q, 16 \times Q\}$. We report *Pass@32*, *Mean@32*, and *Maj@32* metrics, capturing the accuracy and consensus of multiple rollouts. We summarize the results in Table 1. Overall, adding VIP yields consistent improvements on *Pass@32* and *Mean@32* across all three base models and both budgets. For example, on Qwen2.5-Math-1.5B at $8 \times Q$, RLOO$_{+\text{VIP}}$ improves *Pass@32* by +12.3 and *Mean@32* by +6.3 points over RLOO. Similar patterns hold for Llama-3.2-3B-Instruct and Qwen2.5-Math-7B.

We see that the relative performance gain from VIP is larger for the 1.5B and 3B models than for the 7B model. This suggests that VIP's budget-aware variance reduction may particularly help weaker backbones that otherwise underutilize the rollout budget.

**Tool-Augmented Reasoning Task.** We implement our allocation strategy to teach LMs to use a retrieval tool during generation. We follow Chen et al. (2025b) and train on MuSiQue (19,938 prompts) (Trivedi et al., 2022), evaluating on the Bamboogle benchmark (Press et al., 2022) and MuSiQue test set. During training we use `intfloat/e5-base-v2` embeddings (Wang et al., 2022) for the retrieval model. We choose `Qwen2.5-3B-Instruct` as our base model for this experiment. We use *Precision@5* and *F1@5* to evaluate retrieval quality, and *Exact Match (EM)* for final generation correctness. We summarize the results in Table 2.

Under a fixed rollout budget, VIP improves both answer accuracy and retrieval quality in a *coupled* manner. On Bamboogle, Dr. GRPO$_{+\text{VIP}}$ raises EM from 20 to 23.2 while simultaneously lifting F1@5/Precision@5 by $+0.051/+0.060$, and RLOO$_{+\text{VIP}}$ shows larger absolute gains (EM $10.4 \rightarrow 17.6$, F1@5 $0.190 \rightarrow 0.264$, Precision@5 $0.225 \rightarrow 0.294$). The parallel improvements in F1@5 and Precision@5 suggest fewer false positives and better ranking of useful contexts, while EM gains indicate that retrieved evidence is more reliably integrated into final answers. These patterns support VIP as a sample-efficient, domain-agnostic mechanism for tool-augmented reasoning.

Table 1: Percentage results on AIME24 and AIME25. The upper block uses a total rollout budget of $C = 8 \times Q$, and the lower block uses $C = 16 \times Q$. For each pair (Dr. GRPO vs Dr. GRPO$_{+\text{VIP}}$ and RLOO vs RLOO$_{+\text{VIP}}$), higher values are highlighted in green.

| Model | Method | AIME24 | | | AIME25 | | |
|---|---|---|---|---|---|---|---|
| | | Pass@32 | Mean@32 | Maj@32 | Pass@32 | Mean@32 | Maj@32 |
| Qwen2.5-Math-1.5B | Dr. GRPO | 34.56 | 7.91 | 11.26 | 24.03 | 7.62 | 4.89 |
| | **Dr. GRPO+VIP** | 33.63 | 9.167 | 15.5 | 29.82 | 6.14 | 10.76 |
| | RLOO | 18.29 | 3.43 | 6.88 | 15.90 | 2.29 | 8.04 |
| | **RLOO+VIP** | 30.55 | 9.68 | 15.65 | 26.54 | 6.35 | 13.72 |
| Qwen2.5-Math-7B | Dr. GRPO | 50.0 | 19.0 | 34.0 | 34.0 | 10.0 | 15.0 |
| | **Dr. GRPO+VIP** | 58.98 | 23.65 | 38.15 | 36.08 | 10.0 | 19.74 |
| | RLOO | 53.0 | 18.0 | 24.0 | 45.0 | 11.0 | 16.0 |
| | **RLOO+VIP** | 58.0 | 20.0 | 35.0 | 34.0 | 11.0 | 18.0 |
| Llama-3.2-3B-Instruct | Dr. GRPO | 24.68 | 6.25 | 10.62 | 4.28 | 0.21 | 0.07 |
| | **Dr. GRPO+VIP** | 29.18 | 8.85 | 17.21 | 9.37 | 0.94 | 2.35 |
| | RLOO | 28.84 | 8.229 | 13.95 | 9.157 | 0.5208 | 0.3633 |
| | **RLOO+VIP** | 35.59 | 9.479 | 19.0 | 9.99 | 0.625 | 0.61 |
| Qwen2.5-Math-1.5B | Dr. GRPO | 23.96 | 9.479 | 13.64 | 28.21 | 5.83 | 9.97 |
| | **Dr. GRPO+VIP** | 27.37 | 12.08 | 15.01 | 33.05 | 8.43 | 13.77 |
| | RLOO | 18.0 | 3.0 | 7.0 | 22.0 | 2.0 | 3.0 |
| | **RLOO+VIP** | 27.0 | 6.0 | 14.0 | 26.0 | 8.0 | 4.0 |
| Qwen2.5-Math-7B | Dr. GRPO | 52.0 | 19.0 | 35.0 | 36.0 | 10.0 | 20.0 |
| | **Dr. GRPO+VIP** | 57.0 | 20.0 | 33.0 | 36.0 | 13.0 | 21.0 |
| | RLOO | 37.43 | 14.79 | 21.93 | 41.51 | 10.0 | 19.44 |
| | **RLOO+VIP** | 53.85 | 21.77 | 35.59 | 38.32 | 10.31 | 21.4 |
| Llama-3.2-3B-Instruct | Dr. GRPO | 23.26 | 6.35 | 10.09 | 5.49 | 0.63 | 1.31 |
| | **Dr. GRPO+VIP** | 32.34 | 8.23 | 14.96 | 10.67 | 0.52 | 0.05 |
| | RLOO | 29.3 | 6.88 | 1.74 | 9.57 | 0.73 | 1.68 |
| | **RLOO+VIP** | 31.32 | 7.71 | 1.77 | 11.42 | 0.63 | 0.32 |

Table 2: Performance on Bamboogle and MuSiQue. Green cells indicate improvements of the +VIP variant over its base method.

| Method | Bamboogle | | | MuSiQue | | |
|---|---|---|---|---|---|---|
| | EM | F1@5 | Precision@5 | EM | F1@5 | Precision@5 |
| Dr. GRPO | 20 | 0.282 | 0.293 | 6 | 0.123 | 0.126 |
| **Dr. GRPO+VIP** | 23.2 | 0.333 | 0.353 | 10.5 | 0.214 | 0.225 |
| RLOO | 10.4 | 0.190 | 0.225 | 8.5 | 0.178 | 0.179 |
| **RLOO+VIP** | 17.6 | 0.264 | 0.294 | 11 | 0.209 | 0.211 |

Table 3: Ablation study on AIME24 and AIME25. All values are percentages. For each metric, the highest value across methods is highlighted in green.

| Method | AIME24 | | | AIME25 | | |
|---|---|---|---|---|---|---|
| | Pass@32 | Mean@32 | Maj@32 | Pass@32 | Mean@32 | Maj@32 |
| RLOO | 18.29 | 3.43 | 6.88 | 15.90 | 2.29 | 8.04 |
| RLOO+GP+Inverse Acc | 24.61 | 6.77 | 7.90 | 19.10 | 3.33 | 5.03 |
| RLOO+GP+Inverse Var | 25.83 | 4.90 | 10.90 | 21.11 | 2.08 | 3.79 |
| RLOO+Ridge+Allocation | 29.90 | 6.97 | 14.79 | 24.85 | 5.20 | 12.12 |
| **RLOO+VIP** | 30.55 | 9.68 | 15.65 | 26.54 | 6.35 | 13.72 |

**Ablation Studies.** We dissect VIP by ablating its two core components: (i) the *variance predictor* and (ii) the *adaptive budget allocator*. We replace the Gaussian process predictor (Section 5.1) with a Ridge Regression baseline and substitute the adaptive allocator (Section 5.2) with two heuristics, *Inverse Accuracy* and *Inverse Variance* (details in the Appendix). Ablation results on Qwen2.5-Math-1.5B are summarized in Table 3. RLOO+VIP achieves the best performance across metrics, outperforming every ablated variants. Among components, *adaptive allocation* is the most critical: replacing it with heuristics yields substantial drops on all metrics. Substituting the Gaussian process with Ridge Regression produces milder but consistent degradations, showing that calibrated uncertainty further boosts allocation effectiveness. Overall, these results confirm that VIP benefits primarily from variance-aware budget allocation.

**Runtime comparison.** We report the average runtime of each component of our method during a single gradient step in Table 4. The total computational overhead is extremely small: our method adds only 1.12% and 0.83% to the overall RL training time for the 1.5B-parameter and 7B-parameter models, respectively, when including computations cached before training. When excluding cached computations, the overhead further reduces to 0.79% and 0.58%. Because model-forward and rollout costs dominate at larger scales, this relative overhead decreases as model size grows.

Table 4: Wall-clock runtime of core computational components and model-specific operations for Qwen2.5-Math-1.5B and Qwen2.5-Math-7B, measured on a single GPU.

| Model | Operation | Time (s) |
|---|---|---|
| — | Kernel matrix computation (Algorithm 1)* | 10.652 |
| — | Gaussian process training and prediction (Algorithm 1) | 0.745 |
| — | Rollout allocation (Section 5.2 + Algorithm 2) | 29.956 |
| Qwen2.5-Math-1.5B | Rollout sampling | 2781.2 |
| Qwen2.5-Math-1.5B | Log probability computation | 627.68 |
| Qwen2.5-Math-1.5B | Policy update | 263.9 |
| Qwen2.5-Math-7B | Rollout sampling | 3134.4 |
| Qwen2.5-Math-7B | Log probability computation | 1388.8 |
| Qwen2.5-Math-7B | Policy update | 713.5 |

*Cached before training.

**Success probability prediction quality.** To evaluate the quality of our Gaussian Process predictor, we construct a time-series dataset by training Qwen2.5-Math-1.5B and Qwen2.5-Math-7B with RLOO for 55 gradient steps and logging per-prompt success probability at every step. Because the policy changes after each update, the underlying success probabilities drift over time, requiring any predictor to adapt to this non-stationarity. We compare VIP's GP predictor against two baselines, Moving Average and Ridge Regression, using 1024 most recent samples to predict the next mini-batch's success probability. Figure 3 reports the MAE across steps. While the Moving Average and Ridge Regression baselines struggle to track rapid changes in model's behavior, the GP maintains consistently lower MAE throughout training for both model sizes. This indicates that the GP provides a more accurate and adaptive estimator under the non-stationary dynamics of RL training.

## 7 CONCLUSION

This paper introduced Variance-Informed Predictive allocation, a framework for minimizing gradient variance in group-based reinforcement learning. By combining Gaussian process–based predictions of prompt success probabilities with an optimization approach to rollout allocation, VIP uses limited sampling budgets more efficiently. Our experiments on mathematical reasoning and tool-augmented benchmarks demonstrated consistent gains over heuristic or uniform strategies. VIP is a step toward more adaptive, resource-efficient, and principled training pipelines for large language models. Future work will explore its integration with non-verifiable or noisy rewards, opening avenues for its application in RLHF and other alignment paradigms.

**Ethics Statement.** All authors read and agree to adhere to the ICLR Code of Ethics throughout the development, submission, and potential publication of this work. Our research does not involve human subjects, personal data, or any sensitive information. All datasets used are publicly available and do not contain personally identifiable information. We have taken care to ensure that our methods do not introduce or amplify bias, discrimination, or unfairness, as the reward signals in RLVR are determined by rule-based verifiers rather than subjective human judgment. We release all code and experimental protocols to promote transparency and reproducibility. We are not aware of any potential conflicts of interest, legal compliance issues, or research integrity concerns associated with this work.

**Reproducibility Statement.** We have made every effort to ensure that our results are reproducible. All of the models, algorithms, and experiments described in the paper are accompanied by detailed explanations in the main text and appendix. To further support reproducibility, we have provided an anonymized repository containing the full source code, along with instructions for running the experiments and replicating our results. The datasets we used are publicly available, and we include clear descriptions of any preprocessing steps in the appendix. For all theoretical results, we have stated our assumptions explicitly and included complete proofs in the supplementary material.

**LLM Usage Statement.** In preparing this paper, we used large language models as general-purpose tools for tasks such as proofreading, rephrasing, and checking the clarity of some sections. The research ideas, experiments, analysis, and the main writing were carried out by the authors.

**Acknowledgments.** Viet Anh Nguyen gratefully acknowledges the support from the CUHK's Improvement on Competitiveness in Hiring New Faculties Funding Scheme, UGC ECS Grant 24210924, and UGC GRF Grant 14208625. The authors would like the thank Dongxuan Zhu and Hongzheng Yang for their helpful discussion regarding Resource Allocation and Reinforcement Learning.

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

# A PROOFS OF TECHNICAL RESULTS

## A.1 PROOFS OF SECTION 4

*Proof of Proposition 4.2.* We recall the expression of $\tilde{G}_q$:

$$\tilde{G}_q = \frac{1}{n} \sum_{j=1}^{n} (\tilde{R}_j - \frac{1}{n} \sum_{k=1}^{n} \tilde{R}_k) \tilde{Z}_j.$$

The expectation of $\tilde{G}_q$ is:

$$
\begin{aligned}
\mathbb{E}[\tilde{G}_q] &= \mathbb{E}[\frac{1}{n} \sum_{j=1}^{n} (\tilde{R}_j - \frac{1}{n} \sum_{k=1}^{n} \tilde{R}_k) \tilde{Z}_j] \\
&= \mathbb{E}[\frac{1}{n} \sum_{j=1}^{n} \tilde{R}_j \tilde{Z}_j - \frac{1}{n^2} \sum_{j=1}^{n} \sum_{k=1}^{n} \tilde{R}_k \tilde{Z}_j] \\
&= \frac{1}{n} \sum_{j=1}^{n} \mathbb{E}[\tilde{R}_j \tilde{Z}_j] - \frac{1}{n^2} \sum_{j=1}^{n} \sum_{k=1}^{n} \mathbb{E}[\tilde{R}_k \tilde{Z}_j] \\
&= \mathbb{E}[\tilde{R}\tilde{Z}] - \mathbb{E}[\tilde{R}\tilde{Z}] = 0 \qquad\qquad \text{(by Assumption 4.1}(i)\text{)}.
\end{aligned}
$$

The variance of $\tilde{G}_q$ is:

$$
\begin{aligned}
&\text{Var}(\tilde{G}_q) \\
&= \mathbb{E}[\tilde{G}_q^2] - (\mathbb{E}[\tilde{G}_q])^2 = \mathbb{E}[\tilde{G}_q^2] - 0 = \mathbb{E}[\tilde{G}_q^2] \\
&= \mathbb{E}\left[ \left( \frac{1}{n} \sum_{j=1}^{n} \tilde{R}_j \tilde{Z}_j - \frac{1}{n^2} \sum_{j=1}^{n} \sum_{k=1}^{n} \tilde{R}_k \tilde{Z}_j \right)^2 \right] \\
&= \mathbb{E}\left[ \frac{1}{n^2} \left( \sum_{j=1}^{n} \tilde{R}_j \tilde{Z}_j \right)^2 + \frac{1}{n^4} \left( \sum_{j=1}^{n} \sum_{k=1}^{n} \tilde{R}_k \tilde{Z}_j \right)^2 - \frac{2}{n^3} \left( \sum_{j=1}^{n} \tilde{R}_j \tilde{Z}_j \right) \left( \sum_{j=1}^{n} \sum_{k=1}^{n} \tilde{R}_k \tilde{Z}_j \right) \right] \\
&= \underbrace{\mathbb{E}\left[ \frac{1}{n^2} \left( \sum_{j=1}^{n} \tilde{R}_j \tilde{Z}_j \right)^2 \right]}_{\triangleq C_1} + \underbrace{\mathbb{E}\left[ \frac{1}{n^4} \left( \sum_{j=1}^{n} \sum_{k=1}^{n} \tilde{R}_k \tilde{Z}_j \right)^2 \right]}_{\triangleq C_2} - \underbrace{\mathbb{E}\left[ \frac{2}{n^3} \left( \sum_{j=1}^{n} \tilde{R}_j \tilde{Z}_j \right) \left( \sum_{j=1}^{n} \sum_{k=1}^{n} \tilde{R}_k \tilde{Z}_j \right) \right]}_{\triangleq C_3}.
\end{aligned}
$$

Next we compute each term. We find for $C_1$:

$$
C_1 = \mathbb{E}\left[\frac{1}{n^2}\left(\sum_{j=1}^{n}\tilde{R}_j\tilde{Z}_j\right)^2\right] = \frac{1}{n^2}\mathbb{E}\left[\left(\sum_{j=1}^{n}\tilde{R}_j\tilde{Z}_j\right)^2\right]
$$

$$
= \frac{1}{n^2}\mathbb{E}\left[\sum_{j=1}^{n}\tilde{R}_j^2\tilde{Z}_j^2 + 2\sum_{1\leq j<k\leq n}\tilde{R}_j\tilde{Z}_j\tilde{R}_k\tilde{Z}_k\right]
$$

$$
= \frac{1}{n^2}\left(\sum_{j=1}^{n}\mathbb{E}\left[\tilde{R}_j^2\tilde{Z}_j^2\right] + 2\sum_{1\leq j<k\leq n}\mathbb{E}\left[\tilde{R}_j\tilde{Z}_j\tilde{R}_k\tilde{Z}_k\right]\right)
$$

$$
= \frac{1}{n^2}\left(\sum_{j=1}^{n}\mathbb{E}\left[\tilde{R}_j^2\tilde{Z}_j^2\right] + 2\sum_{1\leq j<k\leq n}\mathbb{E}\left[\tilde{R}_j\tilde{Z}_j\right]\mathbb{E}\left[\tilde{R}_k\tilde{Z}_k\right]\right) \quad \text{(independence between } j \text{ and } k)
$$

$$
= \frac{1}{n^2}\left(n\mathbb{E}\left[\tilde{R}^2\tilde{Z}^2\right] + n(n-1)(\mathbb{E}[\tilde{R}\tilde{Z}])^2\right) \quad \text{(identically distributed)}
$$

$$
= \frac{1}{n}\mathbb{E}\left[\tilde{R}^2\tilde{Z}^2\right] + \frac{(n-1)}{n}(\mathbb{E}[\tilde{R}\tilde{Z}])^2
$$

$$
= \frac{1}{n}\mathbb{E}[\tilde{R}^2](\mu_Z^2 + \sigma_Z^2) + \frac{n-1}{n}\mu_z^2(\mathbb{E}[\tilde{R}])^2
$$

$$
= \mathbb{E}[\tilde{R}^2](\frac{1}{n}\mu_Z^2 + \frac{1}{n}\sigma_Z^2) + (\mathbb{E}[\tilde{R}])^2(\frac{n-1}{n}\mu_z^2).
$$

We next compute for $C_2$:

$$
C_2 = \mathbb{E}\left[\frac{1}{n^4}\left(\sum_{j=1}^{n}\sum_{k=1}^{n}\tilde{R}_k\tilde{Z}_j\right)^2\right] = \frac{1}{n^4}\mathbb{E}\left[\sum_{k=1}^{n}\sum_{k'=1}^{n}\sum_{j=1}^{n}\sum_{j'=1}^{n}\tilde{R}_k\tilde{R}_{k'}\tilde{Z}_j\tilde{Z}_{j'}\right].
$$

We can decompose the quadruple sum by whether the indices are equal or not. There are four index-pattern types:

- When $k = k'$ and $j = j'$: there are $n^2$ such terms, each term is $\mathbb{E}[\tilde{R}^2\tilde{Z}^2] = (\mu_Z^2 + \sigma_Z^2)\mathbb{E}[\tilde{R}^2]$. Total contribution to $C_2$ is

$$
T_1 = n^2(\mu_Z^2 + \sigma_Z^2)\mathbb{E}[\tilde{R}^2].
$$

- When $k = k'$ and $j \neq j'$: there are $n^2(n-1)$ such terms. For any fixed $k$ and distinct $j, j'$, $\mathbb{E}[\tilde{R}_k^2\tilde{Z}_j\tilde{Z}_{j'}] = \mathbb{E}[\tilde{R}_k^2]\mathbb{E}[\tilde{Z}_j\tilde{Z}_{j'}] = \mu_Z^2\mathbb{E}[\tilde{R}^2]$. Total contribution to $C_2$ is

$$
T_2 = n^2(n-1)\mu_Z^2\mathbb{E}[\tilde{R}^2].
$$

- When $k \neq k'$ and $j = j'$: there are $n^2(n-1)$ such terms. For distinct $k, k'$, $\mathbb{E}[\tilde{R}_k\tilde{R}_{k'}\tilde{Z}^2] = \mathbb{E}[\tilde{R}_k\tilde{R}_{k'}]\mathbb{E}[\tilde{Z}^2] = (\mu_Z^2 + \sigma_Z^2)(\mathbb{E}[\tilde{R}])^2$. Total contribution to $C_2$ is

$$
T_3 = n^2(n-1)(\mu_Z^2 + \sigma_Z^2)(\mathbb{E}[\tilde{R}])^2.
$$

- When $k \neq k'$ and $j \neq j'$: there are $n^2(n-1)^2$ such terms. For all indices different, $\mathbb{E}[\tilde{R}_k\tilde{R}_{k'}\tilde{Z}_j\tilde{Z}_{j'}] = \mu_Z^2(\mathbb{E}[\tilde{R}])^2$. Total contribution to $C_2$ is

$$
T_4 = n^2(n-1)^2\mu_Z^2(\mathbb{E}[\tilde{R}])^2.
$$

Therefore, we find

$$
\begin{aligned}
C_2 &= \frac{1}{n^4}(T_1 + T_2 + T_3 + T_4) \\
&= \frac{1}{n^4}\Big[ n^2(\mu_Z^2 + \sigma_Z^2)\mathbb{E}[\tilde{R}^2] + n^2(n-1)\mu_Z^2\mathbb{E}[\tilde{R}^2] \\
&\qquad + n^2(n-1)(\mu_Z^2 + \sigma_Z^2)(\mathbb{E}[\tilde{R}])^2 + n^2(n-1)^2\mu_Z^2(\mathbb{E}[\tilde{R}])^2 \Big] \\
&= \mathbb{E}[\tilde{R}^2]\left( \frac{\mu_Z^2}{n} + \frac{\sigma_Z^2}{n^2} \right) + (\mathbb{E}[\tilde{R}])^2\left( \frac{n-1}{n}\mu_Z^2 + \frac{n-1}{n^2}\sigma_Z^2 \right).
\end{aligned}
$$

We next compute for $C_3$:

$$
C_3 = \mathbb{E}\left[ \frac{2}{n^3}\left( \sum_{j=1}^{n} \tilde{R}_j \tilde{Z}_j \right)\left( \sum_{j'=1}^{n}\sum_{k=1}^{n} \tilde{R}_k \tilde{Z}_{j'} \right) \right] = \frac{2}{n^3}\mathbb{E}\Big[ \sum_{j=1}^{n}\sum_{j'=1}^{n}\sum_{k=1}^{n} \tilde{R}_j \tilde{Z}_j \tilde{R}_k \tilde{Z}_{j'} \Big].
$$

We can decompose the triplet sum by whether the indies are equal or not. There are five index-pattern types:

- When $j = k = j'$: there are $n$ such terms. For each $j$, $\mathbb{E}[\tilde{R}_j^2 \tilde{Z}_j^2] = \mathbb{E}[\tilde{R}_j^2]\mathbb{E}[\tilde{Z}_j^2] = (\mu_Z^2 + \sigma_Z^2)\mathbb{E}[\tilde{R}^2]$. Total contribution:
$$
T_1 = n(\mu_Z^2 + \sigma_Z^2)\mathbb{E}[\tilde{R}^2].
$$

- When $j = k \neq j'$: there are $n(n-1)$ such terms. For each $j$ and $j' \neq j$, $\mathbb{E}[\tilde{R}_j^2 \tilde{Z}_j \tilde{Z}_{j'}] = \mathbb{E}[\tilde{R}_j^2]\mathbb{E}[\tilde{Z}_j \tilde{Z}_{j'}] = \mathbb{E}[\tilde{R}^2]\mathbb{E}[\tilde{Z}_j]\mathbb{E}[\tilde{Z}_{j'}] = \mathbb{E}[\tilde{R}^2]\mu_Z^2$. Total contribution is
$$
T_2 = n(n-1)\mathbb{E}[\tilde{R}^2]\mu_Z^2.
$$

- When $j = j' \neq k$: there are $n(n-1)$ such terms. For each $j$ and $k \neq j$, $\mathbb{E}[\tilde{R}_j \tilde{Z}_j^2 \tilde{R}_k] = \mathbb{E}[\tilde{R}_j \tilde{R}_k]\mathbb{E}[\tilde{Z}_j^2] = (\mu_Z^2 + \sigma_Z^2)(\mathbb{E}[\tilde{R}])^2$. Total contribution is
$$
T_3 = n(n-1)(\mu_Z^2 + \sigma_Z^2)(\mathbb{E}[\tilde{R}])^2.
$$

- When $k = j' \neq j$: there are $n(n-1)$ such terms. For each $j$ and $k \neq j$, $\mathbb{E}[\tilde{R}_j \tilde{Z}_j \tilde{R}_k \tilde{Z}_k] = \mathbb{E}[\tilde{R}_j \tilde{R}_k]\mathbb{E}[\tilde{Z}_j \tilde{Z}_k] = \mu_Z^2(\mathbb{E}[\tilde{R}])^2$. Total contribution is
$$
T_4 = n(n-1)\mu_Z^2(\mathbb{E}[\tilde{R}])^2.
$$

- When $j, j', k$ are all distinct: there are $n(n-1)(n-2)$ such terms. For each triple of distinct indices, $\mathbb{E}[\tilde{R}_j \tilde{Z}_j \tilde{R}_k \tilde{Z}_{j'}] = \mu_Z^2(\mathbb{E}[\tilde{R}])^2$. Total contribution is
$$
T_5 = n(n-1)(n-2)\mu_Z^2(\mathbb{E}[\tilde{R}])^2.
$$

Therefore, we find

$$
\begin{aligned}
C_3 &= \frac{2}{n^3}\left( T_1 + T_2 + T_3 + T_4 + T_5 \right) \\
&= \frac{2}{n^3}\Bigg[ n(\mu_Z^2 + \sigma_Z^2)\mathbb{E}[\tilde{R}^2] + n(n-1)\mathbb{E}[\tilde{R}^2]\mu_Z^2 \\
&\qquad + n(n-1)(\mu_Z^2 + \sigma_Z^2)\left( \mathbb{E}[\tilde{R}] \right)^2 + n(n-1)\mu_Z^2\left( \mathbb{E}[\tilde{R}] \right)^2 \\
&\qquad + n(n-1)(n-2)\mu_Z^2\left( \mathbb{E}[\tilde{R}] \right)^2 \Bigg] \\
&= \mathbb{E}[\tilde{R}^2]\left( \frac{2}{n}\mu_Z^2 + \frac{2}{n^2}\sigma_Z^2 \right) + (\mathbb{E}[\tilde{R}])^2\left( \frac{2n-2}{n}\mu_Z^2 + \frac{2n-2)}{n^2}\sigma_Z^2 \right).
\end{aligned}
$$

Group terms with $\mathbb{E}[\tilde{R}^2]$ and $(\mathbb{E}[\tilde{R}])^2$ coefficients:

$$C_1 + C_2 - C_3 = \mathbb{E}[\tilde{R}^2]\left[\left(\frac{1}{n}\mu_Z^2 + \frac{1}{n}\sigma_Z^2\right) + \left(\frac{\mu_Z^2}{n} + \frac{\sigma_Z^2}{n^2}\right) - \left(\frac{2}{n}\mu_Z^2 + \frac{2}{n^2}\sigma_Z^2\right)\right]$$

$$+ (\mathbb{E}[\tilde{R}])^2\left[\frac{n-1}{n}\mu_Z^2 + \left(\frac{n-1}{n}\mu_Z^2 + \frac{n-1}{n^2}\sigma_Z^2\right) - \left(\frac{2n-2}{n}\mu_Z^2 + \frac{2n-2}{n^2}\sigma_Z^2\right)\right].$$

We simplify each bracket to obtain:

$$\mathrm{Var}(\tilde{G}_q) = C_1 + C_2 - C_3 = \frac{n-1}{n^2}\sigma_Z^2\left(\mathbb{E}[\tilde{R}^2] - (\mathbb{E}[\tilde{R}])^2\right) = \frac{n-1}{n^2}\sigma_Z^2\sigma_R^2.$$

For a given prompt, $\tilde{R}$ takes 1 with probability $p$ and $-1$ with probability $1-p$, leading to its variance of $4p(1-p)$. We obtain the final variance of the per-prompt gradient estimator:

$$\mathrm{Var}(\tilde{G}_q) = \frac{\sigma_Z^2(n-1)}{n^2} \cdot 4p(1-p).$$

This completes the proof. $\qquad\square$

*Proof of Proposition 4.3.* We recall the expression of $\tilde{G}_q$:

$$\tilde{G}_q = \frac{1}{n}\sum_{j=1}^{n}\left(\tilde{R}_j - \frac{1}{n-1}\sum_{\substack{k=1\\k\neq j}}^{n}\tilde{R}_k\right)\tilde{Z}_j.$$

The expectation of $\tilde{G}_q$ is:

$$\mathbb{E}[\tilde{G}_q] = \mathbb{E}\left[\frac{1}{n}\sum_{j=1}^{n}\left(\tilde{R}_j - \frac{1}{n-1}\sum_{\substack{k=1\\k\neq j}}^{n}\tilde{R}_k\right)\tilde{Z}_j\right]$$

$$= \mathbb{E}\left[\frac{1}{n}\sum_{j=1}^{n}\tilde{R}_j\tilde{Z}_j - \frac{1}{n(n-1)}\sum_{j=1}^{n}\sum_{\substack{k=1\\k\neq j}}^{n}\tilde{R}_k\tilde{Z}_j\right]$$

$$= \frac{1}{n}\sum_{j=1}^{n}\mathbb{E}[\tilde{R}_j\tilde{Z}_j] - \frac{1}{n(n-1)}\sum_{j=1}^{n}\sum_{\substack{k=1\\k\neq j}}^{n}\mathbb{E}[\tilde{R}_k\tilde{Z}_j]$$

$$= \mathbb{E}[\tilde{R}\tilde{Z}] - \mathbb{E}[\tilde{R}\tilde{Z}] \qquad\qquad \text{(by Assumption 4.1(i))}$$

$$= 0.$$

The variance of $\tilde{G}_q$ is:

$$\mathrm{Var}(\tilde{G}_q)$$

$$= \mathbb{E}[\tilde{G}_q^2] - (\mathbb{E}[\tilde{G}_q])^2 = \mathbb{E}[\tilde{G}_q^2] - 0 = \mathbb{E}[\tilde{G}_q^2]$$

$$= \mathbb{E}\left[\left(\frac{1}{n}\sum_{j=1}^{n}\tilde{R}_j\tilde{Z}_j - \frac{1}{n(n-1)}\sum_{j=1}^{n}\sum_{\substack{k=1\\k\neq j}}^{n}\tilde{R}_k\tilde{Z}_j\right)^2\right]$$

$$= \mathbb{E}\left[\frac{1}{n^2}\left(\sum_{j=1}^{n}\tilde{R}_j\tilde{Z}_j\right)^2 + \frac{1}{n^2(n-1)^2}\left(\sum_{j=1}^{n}\sum_{\substack{k=1\\k\neq j}}^{n}\tilde{R}_k\tilde{Z}_j\right)^2 - \frac{2}{n^2(n-1)}\left(\sum_{j=1}^{n}\tilde{R}_j\tilde{Z}_j\right)\left(\sum_{j=1}^{n}\sum_{\substack{k=1\\k\neq j}}^{n}\tilde{R}_k\tilde{Z}_j\right)\right]$$

$$= \underbrace{\mathbb{E}\left[\frac{1}{n^2}\left(\sum_{j=1}^{n}\tilde{R}_j\tilde{Z}_j\right)^2\right]}_{\triangleq C_1} + \underbrace{\mathbb{E}\left[\frac{1}{n^2(n-1)^2}\left(\sum_{j=1}^{n}\sum_{\substack{k=1\\k\neq j}}^{n}\tilde{R}_k\tilde{Z}_j\right)^2\right]}_{\triangleq C_2} - \underbrace{\mathbb{E}\left[\frac{2}{n^2(n-1)}\left(\sum_{j=1}^{n}\tilde{R}_j\tilde{Z}_j\right)\left(\sum_{j=1}^{n}\sum_{\substack{k=1\\k\neq j}}^{n}\tilde{R}_k\tilde{Z}_j\right)\right]}_{\triangleq C_3}.$$

The first term $C_1$ is already computed in the proof of Proposition 4.2, and we have:

$$C_1 = \mathbb{E}[\tilde{R}^2](\frac{1}{n}\mu_Z^2 + \frac{1}{n}\sigma_Z^2) + (\mathbb{E}[\tilde{R}])^2(\frac{n-1}{n}\mu_z^2).$$

Next, we consider the term $C_2$:

$$C_2 = \mathbb{E}\left[\frac{1}{n^2(n-1)^2}\left(\sum_{j=1}^{n}\sum_{\substack{k=1\\k\neq j}}^{n}\tilde{R}_k\tilde{Z}_j\right)^2\right]$$

$$= \mathbb{E}\left[\frac{1}{n^2(n-1)^2}\left(\left(\sum_{j=1}^{n}\sum_{k=1}^{n}\tilde{R}_k\tilde{Z}_j\right) - \left(\sum_{j=1}^{n}\tilde{R}_j\tilde{Z}_j\right)\right)^2\right]$$

$$= \frac{1}{n^2(n-1)^2}\left(\mathbb{E}\left[\left(\sum_{j=1}^{n}\sum_{k=1}^{n}\tilde{R}_k\tilde{Z}_j\right)^2\right] - 2\,\mathbb{E}\left[\left(\sum_{j=1}^{n}\sum_{k=1}^{n}\tilde{R}_k\tilde{Z}_j\right)\left(\sum_{j'=1}^{n}\tilde{R}_{j'}\tilde{Z}_{j'}\right)\right] + \mathbb{E}\left[\left(\sum_{j=1}^{n}\tilde{R}_j\tilde{Z}_j\right)^2\right]\right).$$

We can utilize the computation from the proof of Proposition 4.2 to have:

$$\mathbb{E}\left[\frac{1}{n^4}\left(\sum_{j=1}^{n}\sum_{k=1}^{n}\tilde{R}_k\tilde{Z}_j\right)^2\right] = \mathbb{E}[\tilde{R}^2]\left(\frac{\mu_Z^2}{n} + \frac{\sigma_Z^2}{n^2}\right) + (\mathbb{E}[\tilde{R}])^2\left(\frac{n-1}{n}\mu_Z^2 + \frac{n-1}{n^2}\sigma_Z^2\right),$$

$$\mathbb{E}\left[\frac{2}{n^3}\left(\sum_{j=1}^{n}\tilde{R}_j\tilde{Z}_j\right)\left(\sum_{j'=1}^{n}\sum_{k=1}^{n}\tilde{R}_k\tilde{Z}_{j'}\right)\right] = \mathbb{E}[\tilde{R}^2]\left(\frac{2}{n}\mu_Z^2 + \frac{2}{n^2}\sigma_Z^2\right) + (\mathbb{E}[\tilde{R}])^2\left(\frac{2n-2}{n}\mu_Z^2 + \frac{2n-2}{n^2}\sigma_Z^2\right),$$

$$\mathbb{E}\left[\frac{1}{n^2}\left(\sum_{j=1}^{n}\tilde{R}_j\tilde{Z}_j\right)^2\right] = \mathbb{E}[\tilde{R}^2]\left(\frac{1}{n}\mu_Z^2 + \frac{1}{n}\sigma_Z^2\right) + (\mathbb{E}[\tilde{R}])^2\left(\frac{n-1}{n}\mu_Z^2\right).$$

Therefore,

$$C_2 = \mathbb{E}[\tilde{R}^2]\left[\frac{n}{(n-1)^2}\mu_Z^2 + \frac{1}{(n-1)^2}\sigma_Z^2 - \left(\frac{2}{(n-1)^2}\mu_Z^2 + \frac{2}{n(n-1)^2}\sigma_Z^2\right) + \frac{1}{n(n-1)^2}\mu_Z^2 + \frac{1}{n(n-1)^2}\sigma_Z^2\right]$$

$$+ (\mathbb{E}[\tilde{R}])^2\left[\frac{n}{(n-1)}\mu_Z^2 + \frac{1}{(n-1)}\sigma_Z^2 - \left(\frac{2}{n-1}\mu_Z^2 + \frac{2}{n(n-1)}\sigma_Z^2\right) + \frac{1}{n(n-1)}\mu_Z^2\right]$$

$$= \mathbb{E}[\tilde{R}^2]\left[\frac{n^2-2n+1}{n(n-1)^2}\mu_Z^2 + \frac{n-1}{n(n-1)^2}\sigma_Z^2\right] + (\mathbb{E}[\tilde{R}])^2\left[\frac{n^2-2n+1}{n(n-1)}\mu_Z^2 + \frac{n-2}{n(n-1)}\sigma_Z^2\right]$$

$$= \mathbb{E}[\tilde{R}^2]\left[\frac{1}{n}\mu_Z^2 + \frac{1}{n(n-1)}\sigma_Z^2\right] + (\mathbb{E}[\tilde{R}])^2\left[\frac{n-1}{n}\mu_Z^2 + \frac{n-2}{n(n-1)}\sigma_Z^2\right].$$

We compute $C_3$ as follows:

$$C_3 = \mathbb{E}\left[\frac{2}{n^2(n-1)}\left(\sum_{j=1}^{n}\tilde{R}_j\tilde{Z}_j\right)\left(\sum_{j'=1}^{n}\sum_{\substack{k=1\\k\neq j'}}^{n}\tilde{R}_k\tilde{Z}_{j'}\right)\right]$$

$$= \mathbb{E}\left[\frac{2}{n^2(n-1)}\left(\sum_{j=1}^{n}\tilde{R}_j\tilde{Z}_j\right)\left(\sum_{j'=1}^{n}\sum_{k=1}^{n}\tilde{R}_k\tilde{Z}_{j'} - \sum_{j'=1}^{n}\tilde{R}_{j'}\tilde{Z}_{j'}\right)\right]$$

$$= \frac{2}{n^2(n-1)}\left(\mathbb{E}\left[\left(\sum_{j=1}^{n}\tilde{R}_j\tilde{Z}_j\right)\left(\sum_{j'=1}^{n}\sum_{k=1}^{n}\tilde{R}_k\tilde{Z}_{j'}\right)\right] - \mathbb{E}\left[\left(\sum_{j=1}^{n}\tilde{R}_j\tilde{Z}_j\right)\left(\sum_{j'=1}^{n}\tilde{R}_{j'}\tilde{Z}_{j'}\right)\right]\right).$$

We can utilize the computation of $\frac{n^3}{2}C_3$ and $n^2C_1$ from the proof of Proposition 4.2 to have:

$$\mathbb{E}\left[\left(\sum_{j=1}^{n}\tilde{R}_j\tilde{Z}_j\right)\left(\sum_{j'=1}^{n}\sum_{k=1}^{n}\tilde{R}_k\tilde{Z}_{j'}\right)\right] = \mathbb{E}[\tilde{R}^2]\left(n^2\mu_Z^2 + n\sigma_Z^2\right) + (\mathbb{E}[\tilde{R}])^2\left(n^2(n-1)\mu_Z^2 + n(n-1)\sigma_Z^2\right),$$

$$\mathbb{E}\left[\left(\sum_{j=1}^{n}\tilde{R}_j\tilde{Z}_j\right)\left(\sum_{j'=1}^{n}\tilde{R}_{j'}\tilde{Z}_{j'}\right)\right] = \mathbb{E}[\tilde{R}^2](n\mu_Z^2 + n\sigma_Z^2) + (\mathbb{E}[\tilde{R}])^2(n(n-1)\mu_z^2).$$

Plugging these terms to the computation of $C_3$ yields us:

$$C_3 = \frac{2}{n^2(n-1)}\left\{\mathbb{E}[\tilde{R}^2]\left(n^2\mu_Z^2 + n\sigma_Z^2\right) + (\mathbb{E}[\tilde{R}])^2\left(n^2(n-1)\mu_Z^2 + n(n-1)\sigma_Z^2\right)\right.$$

$$\left. - \left[\mathbb{E}[\tilde{R}^2]\left(n\mu_Z^2 + n\sigma_Z^2\right) + (\mathbb{E}[\tilde{R}])^2\left(n(n-1)\mu_Z^2\right)\right]\right\}$$

$$= \mathbb{E}[\tilde{R}^2]\cdot\frac{2(n^2-n)}{n^2(n-1)}\mu_Z^2 + (\mathbb{E}[\tilde{R}])^2\cdot\frac{2n(n-1)}{n^2(n-1)}\left((n-1)\mu_Z^2 + \sigma_Z^2\right)$$

$$= \mathbb{E}[\tilde{R}^2]\left(\frac{2}{n}\mu_Z^2\right) + (\mathbb{E}[\tilde{R}])^2\left(\frac{2n-2}{n}\mu_Z^2 + \frac{2}{n}\sigma_Z^2\right).$$

We have:

$$\mathrm{Var}(\tilde{G}_q) = C_1 + C_2 - C_3 = \mathbb{E}[\tilde{R}^2]\left(\frac{1}{n-1}\sigma_Z^2\right) + (\mathbb{E}[\tilde{R}])^2\left(-\frac{1}{n-1}\sigma_Z^2\right)$$

$$= \frac{\sigma_Z^2}{n-1}(\mathbb{E}[\tilde{R}])^2 - (\mathbb{E}[\tilde{R}])^2)$$

$$= \frac{\sigma_Z^2}{n-1}\mathrm{Var}(\tilde{R}).$$

For a given prompt, $\tilde{R}$ takes 1 with probability $p$ and $-1$ with probability $1-p$, leading to its variance of $4p(1-p)$. We obtain the final variance of the per-prompt gradient estimator:

$$\mathrm{Var}(\tilde{G}_q) = \frac{\sigma_Z^2}{n-1}\cdot 4p(1-p).$$

This completes the proof. $\qquad\qquad\qquad\qquad\qquad\qquad\qquad\qquad\qquad\qquad\qquad\qquad$ $\square$

### A.2 Proofs of Section 5

*Proof of Theorem 5.1.* For clarity and continuity, we restate problem (6) before proceeding with the proof:

$$\begin{aligned}\min\quad & \sum_{q\in\mathcal{B}_t}a_q\frac{n_q-1}{n_q^2}\\ \text{s.t.}\quad & \sum_{q\in\mathcal{B}_t}n_q = C\\ & L \le n_q \le U \quad \forall q\in\mathcal{B}_t.\end{aligned} \qquad (10)$$

Let $V(\{n_q\})$ be the objective function of the above problem. We compute the first and second derivatives of the objective function with respect to each coordinate $n_q$:

$$\frac{\partial V}{\partial n_q} = -a_q\frac{n_q-2}{n_q^3}.$$

Since $n_q \ge L \ge 3$, so for all $q$, $\frac{\partial V}{\partial n_q} < 0$. Thus, $V$ is decreasing with respect to each $n_q$ on the feasible set.

For the second derivatives:

$$\frac{\partial^2 V}{\partial n_q \partial n_{q'}} = 0 \quad \forall q \neq q', \quad \frac{\partial^2 V}{\partial n_q^2} = a_q \frac{2n_q - 6}{n_q^4} \geq 0 \quad \forall q \quad (\text{Since } n_q \geq L \geq 3, \text{ and } a_q \geq 0)$$

Therefore, $V$ is convex and decreasing in each $n_q$ on the feasible set

$$\left\{ n \in \mathbb{R}^B : \sum_{q \in \mathcal{B}_t} n_q = C, \quad L \leq n_q \leq U \; \forall q \right\}.$$

Hence, the minimizer exists and is unique whenever the feasible set is nonempty $BL \leq C \leq BU$.

The Lagrangian function is

$$\mathcal{L} = \sum_{q \in \mathcal{B}_t} a_q \frac{n_q - 1}{n_q^2} + \lambda \left( \sum_{q \in \mathcal{B}_t} n_q - C \right) + \sum_{q \in \mathcal{B}_t} \mu_q (L - n_q) + \sum_{q \in \mathcal{B}_t} \nu_q (n_q - U)$$

where $\lambda \in \mathbb{R}$, and $\mu_q, \nu_q \geq 0$ are Lagrangian multipliers. The KKT conditions are:

$$-a_q \frac{n_q - 2}{n_q^3} + \lambda - \mu_q + \nu_q = 0 \qquad \forall q,$$

$$\mu_q \geq 0, \quad \nu_q \geq 0 \qquad \forall q,$$
$$\mu_q(n_q - L) = 0, \quad \nu_q(n_q - U) = 0 \qquad \forall q,$$
$$L \leq n_q \leq U \qquad \forall q,$$
$$\sum_{q \in \mathcal{B}_t} n_q = C.$$

We consider three cases of $n_q$:

- For each $q$ with $L < n_q < U$, the KKT stationarity condition is

$$\lambda = a_q \frac{n_q - 2}{n_q^3},$$

  where $\lambda$ is the Lagrange multiplier for the sum constraint. Note that the right-hand side is decreasing in $n_q$.

  For $n_q = L$, the right-hand side is $a_q \frac{L-2}{L^3}$, and for $n_q = U$, it is $a_q \frac{U-2}{U^3}$. Therefore, for each $q$ and any $\lambda \in (a_q \frac{U-2}{U^3}, a_q \frac{L-2}{L^3})$, there is at most one solution $n_q$ to $a_q \frac{n_q - 2}{n_q^3} = \lambda$ in the interior $(L, U)$. If $\lambda \geq a_q \frac{L-2}{L^3}$ or $\lambda \leq a_q \frac{U-2}{U^3}$, there is no interior solution, and the optimum for $n_q$ must be at a bound.

- If $n_q = L$, then $\mu_q \geq 0$ and $\nu_q = 0$. According to the KKT condition, we obtain:

$$\lambda = a_q \frac{L-2}{L^3} + \mu_q \geq a_q \frac{L-2}{L^3}.$$

- If $n_q = U$, then $\mu_q = 0$ and $\nu_q \geq 0$. According to the KKT condition, we obtain:

$$\lambda = a_q \frac{U-2}{U^3} - \nu_q \leq a_q \frac{U-2}{U^3}.$$

For a value of $\lambda$, for each coordinate, the KKT solution for $n_q$ is defined as:

$$n_q^\star(\lambda) = \begin{cases} U & \text{if } \lambda \leq a_q \frac{U-2}{U^3}, \\ \text{the unique solution to } \lambda = a_q \frac{n_q - 2}{n_q^3} & \text{if } a_q \frac{U-2}{U^3} < \lambda < a_q \frac{L-2}{L^3}, \\ L & \text{if } \lambda \geq a_q \frac{L-2}{L^3}. \end{cases}$$

The coupling constraint $\sum_{q \in \mathcal{B}_t} n_q = C$ is enforced by selecting $\lambda$ such that

$$S(\lambda) \triangleq \sum_{q \in \mathcal{B}_t} n_q^\star(\lambda) = C.$$

Each $n_q^\star(\lambda)$ is non-increasing in $\lambda$ since $a_q \frac{n_q - 2}{n_q^3}$ is decreasing and the projection preserves monotonicity. Consequently, $S(\lambda)$ is also non-increasing. In particular:

- As $\lambda \to -\infty$, $n_q^\star(\lambda) \to U$, so $S(-\infty) = BU$.

- As $\lambda \to +\infty$, $n_q^\star(\lambda) \to L$, so $S(+\infty) = BL$.

Therefore, for any feasible $C$ with $BL \leq C \leq BU$, there exists a unique $\lambda^\star$ such that $S(\lambda^\star) = C$. Moreover, because $S$ is non-increasing, finding $\lambda^\star$ can be done by bisection. If $C > BU$ or $C < BL$, the problem is infeasible. $\qquad\square$

*Proof of Theorem 5.2.* For clarity and continuity, we restate Problem 8 before proceeding with the proof:

$$\begin{aligned} \min \quad & \sum_{q \in \mathcal{B}_t} a_q \frac{1}{n_q} \\ \text{s.t.} \quad & \sum_{q \in \mathcal{B}_t} n_q = C \\ & L \leq n_q \leq U \quad \forall q \in \mathcal{B}_t \end{aligned} \tag{11}$$

Let $V(\{n_q\})$ be the objective function of the above problem. We compute the first and second derivatives of the objective function with respect to each coordinate $n_q$:

$$\frac{\partial V}{\partial n_q} = -a_q \frac{1}{(n_q - 1)^2}$$

Since $n_q \geq L \geq 3$ and $a_q > 0$, we have $\frac{\partial V}{\partial n_q} \leq 0$ for all $q$. Thus, $V$ is decreasing with respect to each $n_q$ on the feasible set.

For the second derivatives:

$$\frac{\partial^2 V}{\partial n_q \partial n_{q'}} = 0 \quad \forall q \neq q', \qquad \frac{\partial^2 V}{\partial n_q^2} = 2a_q \frac{1}{(n_q - 1)^3} > 0 \quad \forall q$$

Therefore, $V$ is convex and decreasing in each $n_q$ on the feasible set

$$\left\{ n \in \mathbb{R}^B : \sum_{q \in \mathcal{B}_t} n_q = C, \quad L \leq n_q \leq U \right\}.$$

Hence, the minimizer exists and is unique whenever the feasible set is nonempty ($BL \leq C \leq BU$).

The Lagrangian function is

$$\mathcal{L} = \sum_{q \in \mathcal{B}_t} a_q \frac{1}{n_q - 1} + \lambda \left( \sum_{q \in \mathcal{B}_t} n_q - C \right) + \sum_{q \in \mathcal{B}_t} \mu_q (L - n_q) + \sum_{q \in \mathcal{B}_t} \nu_q (n_q - U)$$

where $\lambda \in \mathbb{R}, \mu_q, \nu_q \geq 0$. The KKT conditions are:

$$\begin{aligned} & -a_q \frac{1}{(n_q - 1)^2} + \lambda - \mu_q + \nu_q = 0 && \forall q \\ & \mu_q \geq 0, \quad \nu_q \geq 0 && \forall q \\ & \mu_q (n_q - L) = 0, \quad \nu_q (n_q - U) = 0 && \forall q \\ & L \leq n_q \leq U && \forall q \\ & \sum_{q \in \mathcal{B}_t} n_q = C. \end{aligned}$$

We consider three cases of $n_q$:

- For each $q$ with $L < n_q < U$, the KKT stationarity condition is

$$\lambda = a_q \frac{1}{(n_q - 1)^2},$$

where $\lambda$ is the Lagrange multiplier for the sum constraint. Note that the right-hand side is decreasing in $n_q$ since $n_q \geq L \geq 3$.

For $n_q = L$, the right-hand side is $a_q \frac{1}{(L-1)^2}$, and for $n_q = U$, it is $a_q \frac{1}{(U-1)^2}$. Therefore, for each $q$ and any $\lambda \in (a_q \frac{1}{(U-1)^2}, a_q \frac{1}{(L-1)^2})$, there is one solution $n_q = \sqrt{\frac{a_q}{\lambda}} + 1$ to $a_q \frac{1}{(n_q-1)^2} = \lambda$ in the interior $(L, U)$. If $\lambda \geq a_q \frac{1}{(L-1)^2}$ or $\lambda \leq a_q \frac{1}{(U-1)^2}$, there is no interior solution, and the optimum for $n_q$ must be at a bound.

- If $n_q = L$, then $\mu_q \geq 0$ and $\nu_q = 0$. According to the KKT condition, we obtain:

$$\lambda = a_q \frac{1}{(L-1)^2} + \mu_q \geq a_q \frac{1}{(L-1)^2}.$$

- If $n_q = U$, then $\mu_q = 0$ and $\nu_q \geq 0$. According to the KKT condition, we obtain:

$$\lambda = a_q \frac{1}{(U-1)^2} - \nu_q \leq a_q \frac{1}{(U-1)^2}.$$

For a value of $\lambda$, for each coordinate, the KKT solution for $n_q$ is defined as:

$$n_q^\star(\lambda) = \begin{cases} U & \text{if } \lambda \leq a_q \frac{1}{(U-1)^2}, \\ \sqrt{\frac{a_q}{\lambda}} + 1 & \text{if } a_q \frac{1}{(U-1)^2} < \lambda < a_q \frac{1}{(L-1)^2}, \\ L & \text{if } \lambda \geq a_q \frac{1}{(L-1)^2}. \end{cases}$$

The coupling constraint $\sum_{q \in \mathcal{B}_t} n_q = C$ is enforced by selecting $\lambda$ such that

$$S(\lambda) := \sum_{q \in \mathcal{B}_t} n_q^\star(\lambda) = C.$$

Each $n_q^\star(\lambda)$ is non-increasing in $\lambda$ (since $a_q \frac{1}{(n_q-1)^2}$ is decreasing and the projection preserves monotonicity), so $S(\lambda)$ is also non-increasing. In particular:

- As $\lambda \to -\infty$, $n_q^\star(\lambda) \to U$, so $S(-\infty) = BU$.

- As $\lambda \to +\infty$, $n_q^\star(\lambda) \to L$, so $S(+\infty) = BL$.

Therefore, for any feasible $C$ with $BL \leq C \leq BU$, there exists a unique $\lambda$ such that $S(\lambda) = C$. If $C > BU$ or $C < BL$, the problem is infeasible. $\qquad \square$

## B  STATISTICAL TESTS FOR SECOND-ORDER UNCORRELATION

In this section, we provide statistical tests to validate the assumptions in our paper.

### B.1  FIRST-ORDER CORRELATION TEST VIA FISHER'S METHOD

For each question $q$, consider the two random variables $\tilde{R}_q$ and $\tilde{Z}_q$, with $n$ independent observations

$$\{(\tilde{R}_{q,j}, \tilde{Z}_{q,j})\}_{j=1}^n.$$

**Compute per-question Pearson correlation.** The sample Pearson correlation for question $q$ is

$$\hat{\rho}_q = \frac{\sum_{j=1}^n (\tilde{R}_{q,j} - \bar{\tilde{R}}_q)(\tilde{Z}_{q,j} - \bar{Z}_q)}{\sqrt{\sum_{j=1}^n (\tilde{R}_{q,j} - \bar{R}_q)^2 \sum_{j=1}^n (\tilde{Z}_{q,j} - \bar{Z}_q)^2}},$$

where

$$\bar{R}_q = \frac{1}{n} \sum_{j=1}^{n} \tilde{R}_{q,j}, \qquad \bar{Z}_q = \frac{1}{n} \sum_{j=1}^{n} \tilde{Z}_{q,j}.$$

**Compute per-question $p$-values.** For each question $q$, we test the null hypothesis

$$H_{0,q} : \rho_q = 0.$$

The $p$-value $p_q$ is obtained directly from the standard Pearson correlation test.

**Combine $p$-values across questions using Fisher's method.** Let $Q$ be the total number of questions. Fisher's method combines the per-question $p$-values $\{p_q\}_{q=1}^{Q}$ into a single test statistic:

$$\chi^2_{\text{Fisher}} = -2 \sum_{q=1}^{Q} \ln p_q.$$

Under the global null hypothesis

$$H_0 : \rho_q = 0 \quad \forall q,$$

the statistic $\chi^2_{\text{Fisher}}$ follows a chi-squared distribution with $2Q$ degrees of freedom:

$$\chi^2_{\text{Fisher}} \sim \chi^2_{2Q}.$$

**Global $p$-value and decision rule.** The global $p$-value for testing $H_0$ across all questions is

$$p_{\text{global}} = \Pr\left(\chi^2_{2Q} \geq \chi^2_{\text{Fisher}}\right).$$

Given a significance level $\alpha$ (e.g., $\alpha = 0.05$), we make the following decision:

- If $p_{\text{global}} < \alpha$, we reject the global null hypothesis $H_0$, which indicates that at least some of the correlations $\rho_q$ are significantly different from zero across the questions.

- If $p_{\text{global}} \geq \alpha$, we fail to reject $H_0$, which supports the hypothesis that the correlations $\rho_q$ are zero for all questions at the significance level $\alpha$.

We conduct the correlation test described above on a benchmark of $Q = 600$ questions, each with $n = 16$ independent rollouts. For each question $q$, we compute the Pearson correlation between $\tilde{R}_q$ and $\tilde{Z}_q$, obtain the corresponding $p$-value $p_q$, and aggregate across all questions using Fisher's method to compute the global $p$-value $p_{\text{global}}$.

We evaluate the policy model $\pi_{\theta_t}$ at four checkpoints during training of `Qwen2.5-Math-1.5B`, corresponding to 0.0, 0.5, 1.0 epochs. At each checkpoint, we report the resulting $p_{\text{global}}$ values in Table 5. Since all global $p$-values exceed the chosen significance level $\alpha = 0.05$, we do not reject the null hypothesis, which supports our assumption that the correlations $\rho_q$ are zero across all questions.

| Epoch | Global $p$-value $\tilde{Z}_j = \|H(\tilde{o}_j)\|_2$ |
|:-----:|:-----:|
| 0.0 | 0.7322 |
| 0.5 | 0.1108 |
| 1.0 | 0.2186 |

Table 5: Global $p$-values ($p_{\text{global}}$) across training epochs for `Qwen2.5-Math-1.5B`.

## B.2 FIRST-ORDER CORRELATION TEST VIA EDGINGTON'S METHOD

For each question $q$, let $\hat{\rho}_q$ denote the sample Pearson correlation computed from $n$ independent rollouts, and let $p_q$ be the corresponding two-sided $p$-value for testing the null hypothesis

$$H_{0,q} : \rho_q = 0.$$

To aggregate evidence across all $Q$ questions, we apply Edgington's sum-of-$p$ method.

**Sum of $p$-values.** Each per-question $p_q$ is treated as a realization of a $\mathrm{Uniform}(0, 1)$ variable under its null hypothesis. Edgington's statistic is defined by the simple sum

$$S_{\mathrm{Ed}} = \sum_{q=1}^{Q} p_q.$$

**Null distribution.** Under the global null hypothesis

$$H_0 : \rho_q = 0 \quad \forall q,$$

each $p_q \sim \mathrm{Uniform}(0, 1)$, and therefore

$$S_{\mathrm{Ed}} \sim \mathrm{Irwin\text{–}Hall}(Q),$$

with mean and variance

$$\mathbb{E}[S_{\mathrm{Ed}}] = \frac{Q}{2}, \qquad \mathrm{Var}(S_{\mathrm{Ed}}) = \frac{Q}{12}.$$

For large $Q$, $S_{\mathrm{Ed}}$ is well approximated by a normal distribution:

$$S_{\mathrm{Ed}} \approx \mathcal{N}\left(\frac{Q}{2}, \frac{Q}{12}\right).$$

**Global $p$-value and decision rule.** Small values of $S_{\mathrm{Ed}}$ indicate joint evidence against $H_0$. The corresponding one-sided global $p$-value is

$$p_{\mathrm{global}} = \Phi\left(\frac{S_{\mathrm{Ed}} - Q/2}{\sqrt{Q/12}}\right),$$

where $\Phi$ denotes the standard normal CDF. Given a significance level $\alpha = 0.5$, we reject $H_0$ when $p_{\mathrm{global}} < \alpha$.

We set up the experiment identically to the Fisher's method test in Appendix B.1, using the same benchmark of $Q = 600$ questions, each with $n = 16$ independent rollouts. For each checkpoint of the policy model $\pi_{\theta_t}$, we compute the Edgington statistic and report the global $p$-value. Since all global $p$-values exceed the chosen significance level $\alpha = 0.05$, we do not reject the null hypothesis, which supports our assumption that the correlations $\rho_q$ are zero across all questions.

| Epoch | Global $p$-value |
| --- | --- |
| | $\tilde{Z}_j = \|H(\tilde{o}_j)\|_2$ |
| 0.0 | 0.7894 |
| 0.5 | 0.3964 |
| 1.0 | 0.2148 |

Table 6: Global $p$-values ($p_{\mathrm{global}}$) across training epochs for `Qwen2.5-Math-1.5B` using Edgington's method.

### B.3 EQUAL VARIANCE TEST VIA LEVENE'S TEST

In the numerical experiments, we have assumed that the variance for $\tilde{Z}_q$ is constant across different prompts $q$. We proceed with a hypothesis test:

$$H_0 : \sigma^2_{Z_q} = \sigma^2_{Z_{q'}} \ \forall q \neq q', \qquad H_1 : \text{At least one } \sigma^2_{Z_q} \neq \sigma^2_{Z_{q'}}$$

For each question $q$, consider the random variable $\tilde{Z}_q$ with $n_q$ independent observations $\{\tilde{Z}_{q,j}\}_{j=1}^{n_q}$.

**Transform observations for Levene's test.** Let $Y_{q,j}$ denote the absolute deviation from the per-question median:

$$Y_{q,j} = \left|\tilde{Z}_{q,j} - \mathrm{median}(\tilde{Z}_{q,1}, \ldots, \tilde{Z}_{q,n_q})\right|.$$

**Compute group means of transformed observations.** The mean of the transformed observations for question $q$ is

$$\bar{Y}_q = \frac{1}{n_q} \sum_{j=1}^{n_q} Y_{q,j},$$

and the overall mean across all questions is

$$\bar{Y} = \frac{1}{N} \sum_{q=1}^{Q} \sum_{j=1}^{n_q} Y_{q,j}, \qquad N = \sum_{q=1}^{Q} n_q.$$

**Compute Levene's test statistic.** The test statistic is given by

$$W = \frac{(N-Q) \sum_{q=1}^{Q} n_q (\bar{Y}_q - \bar{Y})^2}{(Q-1) \sum_{q=1}^{Q} \sum_{j=1}^{n_q} (Y_{q,j} - \bar{Y}_q)^2}.$$

Under the null hypothesis that the variances are equal across questions,

$$H_0 : \sigma_{Z_q}^2 = \sigma_{Z_{q'}}^2, \quad \forall q \neq q',$$

the statistic $W$ approximately follows an $F$-distribution with $Q-1$ and $N-Q$ degrees of freedom $W \sim F_{Q-1,N-Q}$.

**Compute $p$-value and decision rule.** The $p$-value for testing $H_0$ is

$$p_{\text{Levene}} = \Pr(F_{Q-1,N-Q} \geq W).$$

Given a significance level $\alpha$ (e.g., $\alpha = 0.05$), we make the following decision:

- If $p_{\text{Levene}} < \alpha$, we reject $H_0$, indicating that the variances of $\tilde{Z}_q$ differ across questions.
- If $p_{\text{Levene}} \geq \alpha$, we fail to reject $H_0$, the hypothesis that the variances are equal across all questions, at the significance level $\alpha$.

We conduct the variance homogeneity test described above on a benchmark of $Q = 600$ questions, each with $n = 16$ independent rollouts. We perform Levene's test across all questions to assess the equality of variances. We evaluate the policy model $\pi_{\theta_t}$ at four checkpoints during training of `Qwen2.5-Math-1.5B`, corresponding to 0.0, 0.5, 1.0 epochs. At each checkpoint, we report the resulting global $p$-values $p_{\text{Levene}}$ in Table 7. Since all $p_{\text{Levene}}$ exceed the chosen significance level $\alpha = 0.05$, we can not reject the null hypothesis, which supports our assumption that the variances $\sigma_{Z_q}^2$ are equal across all questions.

| Epoch | Global $p$-value | |
|-------|-------------------------------------|-------------------------------|
|       | $\tilde{Z}_j = \mathbb{1}^\top H(\tilde{o}_j)$ | $\tilde{Z}_j = \|H(\tilde{o}_j)\|_2$ |
| 0.0   | 0.5019 | 0.2705 |
| 0.5   | 0.4132 | 0.4785 |
| 1.0   | 0.3847 | 0.3847 |

Table 7: $p_{\text{Levene}}$ from Levene's test across training epochs for `Qwen2.5-Math-1.5B`, assessing variance homogeneity of $\tilde{Z}_q$.

### B.4 EQUAL VARIANCE TEST VIA O'BRIEN'S TEST

In the numerical experiments, we have assumed that the variance for $\tilde{Z}_q$ is constant across different prompts $q$. We proceed with a hypothesis test:

$$H_0 : \sigma_{Z_q}^2 = \sigma_{Z_{q'}}^2, \ \forall q \neq q', \qquad H_1 : \text{At least one } \sigma_{Z_q}^2 \neq \sigma_{Z_{q'}}^2$$

For each question $q$, consider the random variable $\tilde{Z}_q$ with $n_q$ independent observations $\{\tilde{Z}_{q,j}\}_{j=1}^{n_q}$.

**Transform observations for O'Brien's test.** Let $Y_{q,j}$ denote O'Brien's transformation of the observations:

$$Y_{q,j} = \frac{(n_q - 1.5)n_q(\tilde{Z}_{q,j} - \bar{\tilde{Z}}_q)^2 - 0.5s_q^2(n_q - 1)}{(n_q - 1)(n_q - 2)},$$

where $\bar{\tilde{Z}}_q$ is the sample mean for question $q$, and $s_q^2$ is the unbiased sample variance for question $q$.

**Compute group means of transformed observations.** The mean of the transformed observations for question $q$ is

$$\bar{Y}_q = \frac{1}{n_q} \sum_{j=1}^{n_q} Y_{q,j},$$

and the overall mean across all questions is

$$\bar{Y} = \frac{1}{N} \sum_{q=1}^{Q} \sum_{j=1}^{n_q} Y_{q,j}, \qquad N = \sum_{q=1}^{Q} n_q.$$

**Compute O'Brien's test statistic.** The test statistic is given by

$$W_{\text{OB}} = \frac{(N - Q) \sum_{q=1}^{Q} n_q (\bar{Y}_q - \bar{Y})^2}{(Q - 1) \sum_{q=1}^{Q} \sum_{j=1}^{n_q} (Y_{q,j} - \bar{Y}_q)^2}.$$

Under the null hypothesis that the variances are equal across questions,

$$H_0 : \sigma_{Z_q}^2 = \sigma_{Z_{q'}}^2 \quad \forall q \neq q',$$

the statistic $W_{\text{OB}}$ approximately follows an $F$-distribution with $Q-1$ and $N-Q$ degrees of freedom $W_{\text{OB}} \sim F_{Q-1,N-Q}$.

**Compute $p$-value and decision rule.** The $p$-value for testing $H_0$ is

$$p_{\text{OB}} = \Pr(F_{Q-1,N-Q} \geq W_{\text{OB}}).$$

Given a significance level $\alpha$ (e.g., $\alpha = 0.05$), we make the following decision:

- If $p_{\text{OB}} < \alpha$, we reject $H_0$, indicating that the variances of $\tilde{Z}_q$ differ across questions.
- If $p_{\text{OB}} \geq \alpha$, we fail to reject $H_0$, the hypothesis that the variances are equal across all questions, at the significance level $\alpha$.

We conduct the variance homogeneity test described above on a benchmark of $Q = 600$ questions, each with $n = 16$ independent rollouts. We perform O'Brien's test across all questions to assess the equality of variances. We evaluate the policy model $\pi_{\theta_t}$ at three checkpoints during training of `Qwen2.5-Math-1.5B`, corresponding to 0.0, 0.5, 1.0 epochs. At each checkpoint, we report the resulting global $p$-values $p_{\text{OB}}$ in Table 8. Since all $p_{\text{OB}}$ exceed the chosen significance level $\alpha = 0.05$, we cannot reject the null hypothesis, which supports our assumption that the variances $\sigma_{Z_q}^2$ are equal across all questions.

| Epoch | Global $p$-value $\tilde{Z}_j = \|H(\tilde{o}_j)\|_2$ |
|:---:|:---:|
| 0.0 | 0.3009 |
| 0.5 | 0.2563 |
| 1.0 | 0.2420 |

Table 8: $p_{\text{OB}}$ from O'Brien's test across training epochs for `Qwen2.5-Math-1.5B`, assessing variance homogeneity of $\tilde{Z}_q$.

## C  ADDITIONAL INFORMATION ON NUMERICAL EXPERIMENTS

**Hyperparameters.** We curate a list of important training hyperparameters for our experiment in Table 9.

Table 9: Hyperparameter configuration.

| Category | Hyperparameter | Value / Setting |
|---|---|---|
| Optimizer | Optimizer | AdamW |
| | Learning rate | $1 \times 10^{-6}$ |
| | Warm-up | 20 rollout steps |
| rollout | Prompt batch size | 512 |
| | Responses per prompt | 6/8/Dynamic |
| Training | Mini-batch size | 512 |
| | Max generation length | 10 240 tokens |
| | Temperature | 1.0 |

### C.1 ADDITIONAL INFORMATION ON ABLATION STUDIES

**Inverse-accuracy allocation.** We allocate more rollout budget to prompts with lower empirical accuracy. Concretely, letting $\mathrm{acc}_i$ denote the running accuracy estimate for prompt $i$, we set target weights $w_i \propto (1 - \mathrm{acc}_i + \epsilon)$ and normalize to meet the global budget and per-prompt bounds.

**Inverse-variance allocation.** We allocate more rollout budget to prompts whose answers exhibit lower variance. Letting $\sigma_i^2$ be the (running) answer variance estimate, we set $w_i \propto 1/(\sigma_i^2 + \epsilon)$ with the same normalization.

Both heuristics are implemented via a continuously relaxed, constrained optimization that enforces the total-budget and box constraints; we solve it with an online solver and then map fractional solutions to integers using the rounding heuristic.

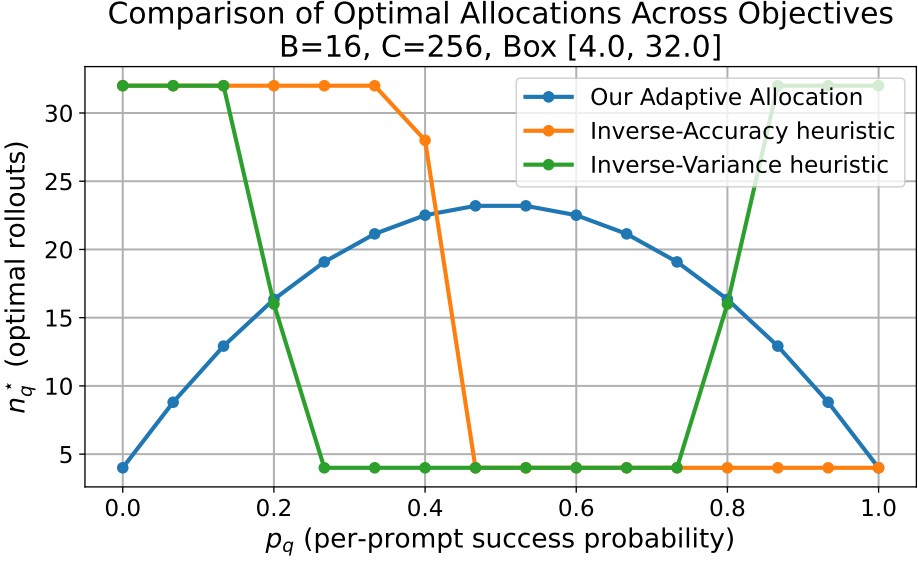

Figure 2: Comparison of optimal rollout allocations produced by different heuristics versus our proposed variance-aware allocation strategy. The figure plots the optimal number of rollouts $n_i^\star$ against prompt difficulty $p_i$, highlighting how our method allocates budget differently from inverse-accuracy and inverse-variance baselines.

### C.2 PROMPT TEMPLATE.

During training, we only use one prompt template for every prompt in the dataset. There are two prompt templates, one for mathematical reasoning and one for tool-augmented reasoning.

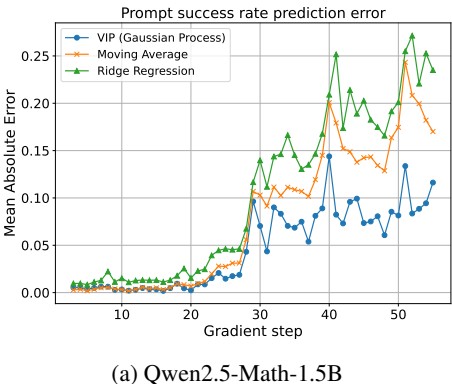

(a) Qwen2.5-Math-1.5B

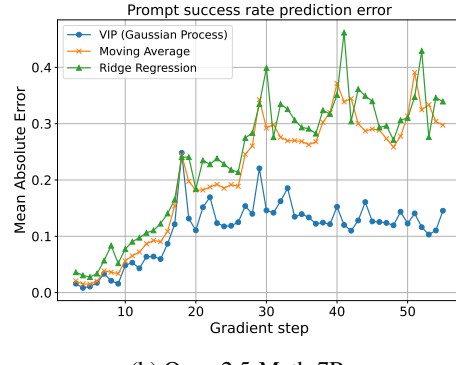

(b) Qwen2.5-Math-7B

Figure 3: Prediction mean absolute error (MAE) over training steps for two model scales. Our GPR-based predictor achieves consistently lower MAE than moving average and Ridge Regression baselines for both the 1.5B and 7B models.

Figure 4: Prompt template for mathematical reasoning

```
Solve the following math problem step by step. The last line of your
    response should be of the form Answer: $Answer (without quotes)
    where $Answer is the answer to the problem. Do not wrap $Answer with
    \boxed{}.

current question: {{question}}

Below are two examples for format reference.
Example question 1: Solve for x: 3x - 5 = 16.

Response:
Add 5 to both sides: 3x = 21.
Divide both sides by 3: x = 7.
Answer: 7

Solve the current question. Remember to put your answer on its own line
    after "Answer:".
```

# D  ALGORITHMS

The algorithm capturing the complete flow the posterior update for the Gaussian Process is provided in Algorithm 1.

Algorithm 2 presents our heuristic rounding procedure, which maps a continuous solution to a discrete one while ensuring that the budget constraints remain satisfied.

# E  EXTENSION TO CONTINUOUS REWARDS

This section details the necessary adaptations to our predictive rollout allocation strategy for the case where the reward $R(\tilde{o}_j)$ is a real-valued random variable. All definitions, assumptions, and notation follow the main text unless otherwise stated.

## E.1  GRADIENT VARIANCE FOR CONTINUOUS REWARDS

We first state the analogues of our variance propositions for the continuous reward setting. The proofs are intermediate results from proofs for binary case in Appendix A.

Figure 5: Prompt template for tool augmented reasoning

```
In this environment you have access to a set of tools you can use to
    assist with the user query.

You may perform multiple rounds of function calls.

In each round, you can call one or more functions.

Here are available functions in JSONSchema format:
    \n```json\n{func_schemas}\n```

In your response, you need to first think about the reasoning process in
    the mind and then conduct function calling to get the information or
    perform the actions if needed. \
The reasoning process and function calling are enclosed within <think>
    </think> and <tool_call> </tool_call> tags. \
The results of the function calls will be given back to you after
    execution, \
and you can continue to call functions until you get the final answer
    for the user's question. \
Finally, if you have got the answer, enclose it within \\boxed{{}} with
    latex format and do not continue to call functions, \
i.e., <think> Based on the response from the function call, I get the
    weather information. </think> The weather in Beijing on 2025-04-01
    is \\[ \\boxed{{20C}} \\].

For each function call, return a json object with function name and
    arguments within <tool_call></tool_call> XML tags:
<tool_call>
{{"name": <function-name>, "arguments": <args-json-object>}}
</tool_call>
```

---

**Algorithm 1** Recursive GP Posterior Update

---

**Require:** Mini-batch $\mathcal{B}_t$; rollout allocation $\{n_q\}_{q=1}^{\mathcal{B}_t}$; prior mean $m_t(\mathcal{D}) \in \mathbb{R}^Q$, kernel matrix $\Sigma \in \mathbb{R}^{Q \times Q}$;

1: **for** each $q \in \mathcal{B}_t$ **do**
2:    # Run $n_q$ rollouts and observe outcomes $\tilde{R}_j \in \{-1, 1\}$
3:    $\bar{R}_q \leftarrow \frac{1}{n_q} \sum_{j=1}^{n_q} \tilde{R}_j$
4:    $\hat{g}_q \leftarrow \text{sigmoid}^{-1}\left(\text{clip}\left(\frac{\bar{R}_q + 1}{2}, \epsilon, 1 - \epsilon\right)\right)$
5: **end for**
6: $g_t^{\text{observe}} \leftarrow (\hat{g}_q)_{q \in \mathcal{B}_t}$
7: Partition $m_t$ and $\Sigma$ according to $\mathcal{B}_t$ and $\mathcal{B}_t^c$
8: $m_{t,\mathcal{B}_t^c}^\star \leftarrow m_{t,\mathcal{B}_t^c} + \Sigma_{\mathcal{B}_t^c \mathcal{B}_t} \Sigma_{\mathcal{B}_t \mathcal{B}_t}^{-1} (g_t^{\text{observe}} - m_{t,\mathcal{B}_t})$
9: $\Sigma^\star \leftarrow \Sigma_{\mathcal{B}_t^c \mathcal{B}_t^c} - \Sigma_{\mathcal{B}_t^c \mathcal{B}_t} \Sigma_{\mathcal{B}_t \mathcal{B}_t}^{-1} \Sigma_{\mathcal{B}_t \mathcal{B}_t^c}$
10: **for** $q = 1$ **to** $Q$ **do**
11:    **if** $q \in \mathcal{B}_t$ **then** $m_{t+1}(x_q) \leftarrow \hat{g}_q$ **else** $m_{t+1}(x_q) \leftarrow m_{t,\mathcal{B}_t^c}^\star(x_q)$ **end if**
12: **end for**
13: $\hat{p}_{t+1} = \text{sigmoid}(m_{t+1}(\mathcal{D}))$
14: **return** $\{\hat{p}_{t+1}\}, m_{t+1}$

---

**Algorithm 2** Heuristic rounding for integer rollout allocation

---

**Require:** Solution $\{n_q^\star\}$, total budget $C$, bounds $\{L, U\}$, objective functions $f_q(\cdot)$ for each $q$

1: For each $q$, set $\hat{n}_q \leftarrow \lfloor n_q^\star \rfloor$
2: $C_{\text{rem}} \leftarrow C - \sum_{q \in \mathcal{B}_t} \hat{n}_q$
3: **for** each $q$ with $\hat{n}_q < U$ **do**
4:    Compute incentive: $\Delta_q \leftarrow f_q(\hat{n}_q) - f_q(\hat{n}_q + 1)$
5: **end for**
6: **while** $C_{\text{rem}} > 0$ **do**
7:    Select $q^\star = \arg\max_{q:\hat{n}_q < U} \Delta_q$
8:    Set $\hat{n}_{q^\star} \leftarrow \hat{n}_{q^\star} + 1$
9:    Recompute $\Delta_{q^\star} \leftarrow f_{q^\star}(\hat{n}_{q^\star}) - f_{q^\star}(\hat{n}_{q^\star} + 1)$
10:    $C_{\text{rem}} \leftarrow C_{\text{rem}} - 1$
11: **end while**
12: **return** Integer allocation $\{\hat{n}_q\}$ with $\sum_{q \in \mathcal{B}_t} \hat{n}_q = C$ and $L \leq \hat{n}_q \leq U$ for all $q$

---

**Proposition E.1** (Dr. GRPO gradient variance, continuous reward). *Let $R(\tilde{o}_j) = \tilde{R}$ be a real-valued random variable with variance $\text{Var}(\tilde{R})$. If Assumption 4.1 holds and $\text{Var}(\tilde{Z}) = \sigma_Z^2$, then the variance of the per-prompt projected Dr. GRPO gradient estimator with $n$ rollouts is*

$$\text{Var}(\tilde{G}) = \frac{(n-1)\sigma_Z^2}{n^2} \text{Var}(\tilde{R}).$$

**Proposition E.2** (RLOO gradient variance, continuous reward). *Let $R(\tilde{o}_j) = \tilde{R}$ be a real-valued random variable with variance $\text{Var}(\tilde{R})$. If Assumption 4.1 holds and $\text{Var}(\tilde{Z}) = \sigma_Z^2$, then the variance of the per-prompt projected RLOO gradient estimator with $n$ rollouts is*

$$\text{Var}(\tilde{G}) = \frac{\sigma_Z^2}{n-1} \text{Var}(\tilde{R}).$$

### E.2 GAUSSIAN PROCESS PREDICTION OF REWARD VARIANCE

For continuous rewards, the per-prompt gradient variance depends on $\text{Var}(\tilde{R}_q)$, which is not directly observable prior to rollout. To predict this quantity, we replace the GP model for success probability with a GP model for reward variance. Specifically, for each prompt $q$, we model the reward variance as $v_{q,t} = \text{softplus}(g_t(x_q)) = \log(1 + \exp(g_t(x_q)))$, where $g_t$ is a latent GP as in the main text. After observing rewards $\{\tilde{R}_{q,j}\}_{j=1}^{n_q}$, we compute the sample variance $\hat{s}_q^2$ and set the observation for

the latent variable as $\hat{g}_{q,t} = \log(\exp(\hat{s}_q^2) - 1)$. The GP posterior update and recursive prediction steps proceed identically, replacing the sigmoid link with the softplus link.

### E.3 BUDGET ALLOCATION OPTIMIZATION

Given predicted reward variances $\widehat{\mathrm{Var}}(\tilde{R}_q)$, we define $a_q := \sigma_{Z_q}^2 \widehat{\mathrm{Var}}(\tilde{R}_q)$. The continuous relaxation of the rollout allocation problem for Dr. GRPO becomes

$$\min\left\{\sum\nolimits_{q \in \mathcal{B}_t} a_q \frac{n_q - 1}{n_q^2} \ : \ \sum\nolimits_{q \in \mathcal{B}_t} n_q = C, \ L \le n_q \le U, \ n_q \in \mathbb{R} \ \forall q\right\},$$

and for RLOO,

$$\min\left\{\sum\nolimits_{q \in \mathcal{B}_t} a_q \frac{1}{n_q - 1} \ : \ \sum\nolimits_{q \in \mathcal{B}_t} n_q = C, \ L \le n_q \le U, \ n_q \in \mathbb{R} \ \forall q\right\}.$$

The optimal solutions are given by Theorems 5.1 and 5.2 in the main text, now with the updated definition of $a_q$. The rounding procedure described in Appendix D applies without modification.

## F TRAINING EVOLUTION COMPARISON

In this section, we compare training evolution on model `Qwen2.5-Math-1.5B` using GRPO and GRPO+VIP. Figures 6 report training metrics (mean advantages, mean rewards) and multiple performance metrics (*best@32*, *maj@32*, *mean@32*).

To ensure that all training trajectories are directly comparable, **every model is trained on the same dataset under identical optimization settings**: the same fixed ordering of 17k training prompts, two epochs of training, a batch size of 512, mini-batch size of 64, and rollout budget per batch of 512 * 8 and 512 * 16. As a result, each gradient step corresponds to the same amount of data and computation across all methods.

Across most of evaluation checkpoints, we observe consistent and pronounced improvements from using VIP. During training, we observe that using VIP, we obtain consistently far-from-zero advantages and higher rewards. This indicates that our learning algorithm provides richer learning signal and our rollout allocation strategy is effective. Further empirical evidence are provided in Figures 7 where VIP performance exceeded GRPO at most training steps.

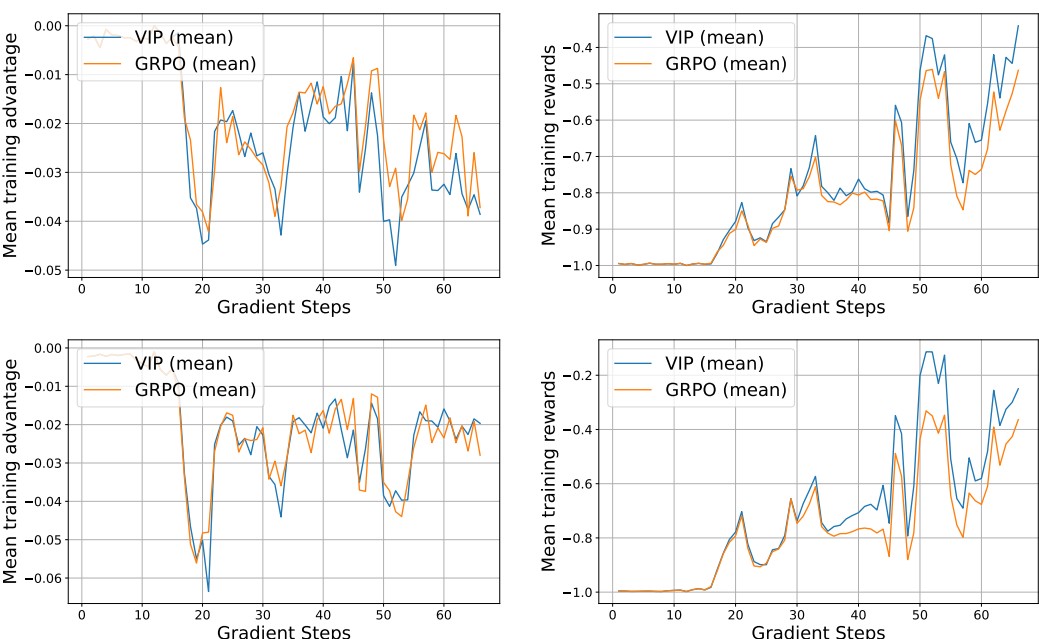

Figure 6: Mean advantages and mean rewards of GRPO vs. GRPO+VIP using rollout budgets of 8 and 16.

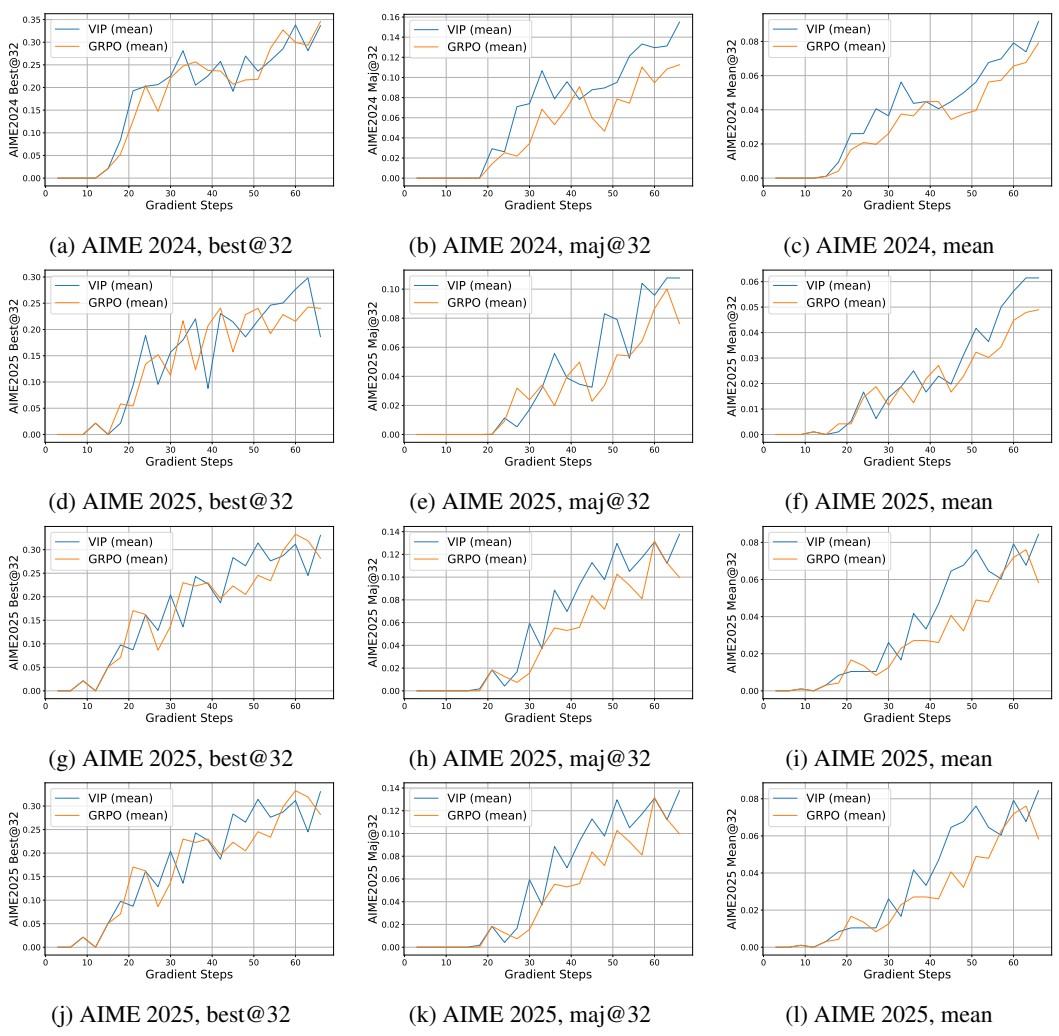

(a) AIME 2024, best@32

(b) AIME 2024, maj@32

(c) AIME 2024, mean

(d) AIME 2025, best@32

(e) AIME 2025, maj@32

(f) AIME 2025, mean

(g) AIME 2025, best@32

(h) AIME 2025, maj@32

(i) AIME 2025, mean

(j) AIME 2025, best@32

(k) AIME 2025, maj@32

(l) AIME 2025, mean

Figure 7: GRPO vs. GRPO+VIP using rollout budget 8 and 16 on AIME 2024 and 2025 across different accuracy metrics.

