In the off-policy regime, importance ratios $r_{j,\tau}(\theta)$ rarely deviate from 1. This indicates that even partially off-policy training methods produce updates that are close to on-policy, a phenomenon particularly pronounced in LLM post-training. Consequently, Assumption 3.1 is unlikely to be restrictive in our setting.

To support this assumption empirically, we measure importance ratios on the response tokens of off-policy samples from our training runs. Prompt and padding tokens are excluded from this analysis. Our evaluation uses 2,560 prompts sampled across different stages of training for `Qwen2.5-Math-1.5B`, with 4 rollouts per prompt. We then collect the importance ratios for all generated tokens and compute the fraction that falls within the interval $[1-\alpha, 1+\alpha]$ for several values of $\alpha$. The results are summarized in Table 10.

These results confirm that the vast majority of importance ratios remain extremely close to 1, providing strong empirical justification for the approximation $r_{j,\tau}(\theta) \approx 1$ in our analysis.

| $\alpha$ | Percentage in $[1-\alpha, 1+\alpha]$ |
|---|---|
| 5e-02 | 97.85% |
| 5e-03 | 82.46% |
| 5e-04 | 71.51% |

Table 10: Fraction of response tokens whose importance ratios fall within $[1-\alpha, 1+\alpha]$ for various choices of $\alpha$.

## G  TRAINING EVOLUTION COMPARISON

In this section, we assess the robustness and stability of our method by retraining `Qwen2.5-Math-1.5B` using GRPO, RLOO, and their VIP-augmented counterparts (GRPO+VIP, RLOO+VIP) across **five random seeds**. Figures 6 and 7 report the mean and standard deviation for multiple performance metrics (*best@32*, *maj@32*, *mean@32*).

To ensure that all training trajectories are directly comparable, **every model is trained on the same dataset under identical optimization settings**: the same fixed ordering of 17k training prompts, one epoch of training, a batch size of 512, mini-batch size of 64, and rollout budget per batch of 512 * 8. As a result, each gradient step corresponds to the same amount of data and computation across all methods.

Across all seeds and evaluation checkpoints, we observe consistent and pronounced improvements from using VIP:

**(i) Faster early-stage learning.** VIP yields substantial gains in the early phase of training. For example, on AIME2024 *mean@32*, RLOO+VIP reaches an accuracy of **0.0316** by step 10, whereas RLOO reaches only **0.0056**—a **6×** **increase**. Similar trends appear in both *best@32* and *maj@32* metrics across AIME2024 and AIME2025.

**(ii) Steeper and more reliable improvement per gradient step.** VIP consistently increases the slope of the learning curve. Its trajectories rise smoothly and monotonically, while the baselines (particularly GRPO on AIME2025 *best@32*) often progress slowly or temporarily plateau between steps 10–20. This shows that variance-aware allocation accelerates the effective learning rate without introducing instability.

**(iii) Increased training stability.** VIP reduces variance across seeds and produces smoother learning curves, reflecting more stable gradient updates. This aligns with the goal of variance-informed allocation: reducing gradient noise directly translates into more predictable and reliable optimization dynamics.

Together, these results demonstrate that VIP improves both the **speed** and the **stability** of GRPO and RLOO training, leading to faster convergence and consistently higher performance throughout the entire training trajectory.

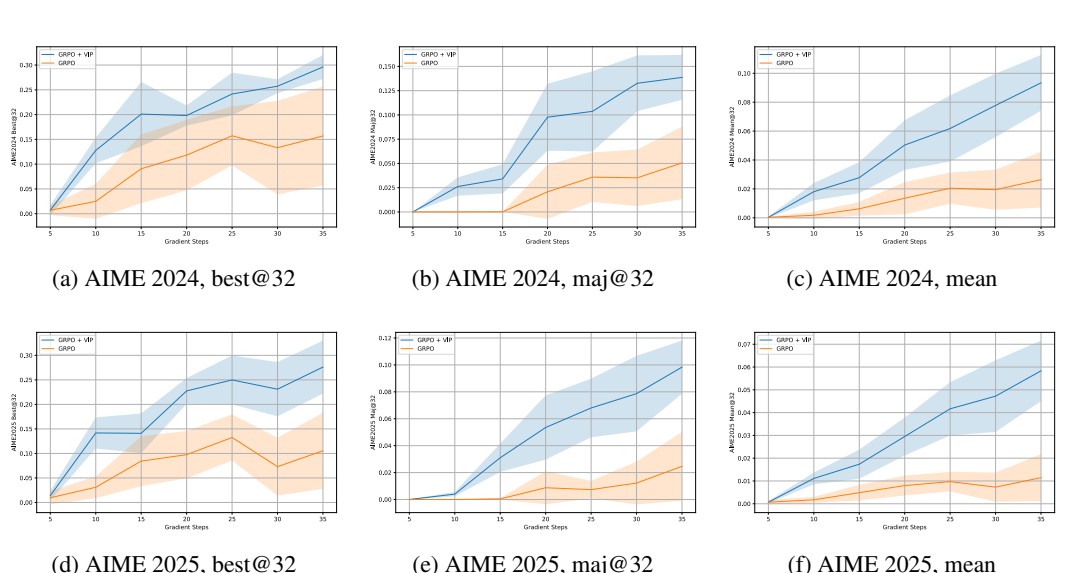

(a) AIME 2024, best@32     (b) AIME 2024, maj@32     (c) AIME 2024, mean

(d) AIME 2025, best@32     (e) AIME 2025, maj@32     (f) AIME 2025, mean

Figure 6: GRPO vs. GRPO+VIP on AIME 2024 and 2025 across different accuracy metrics.

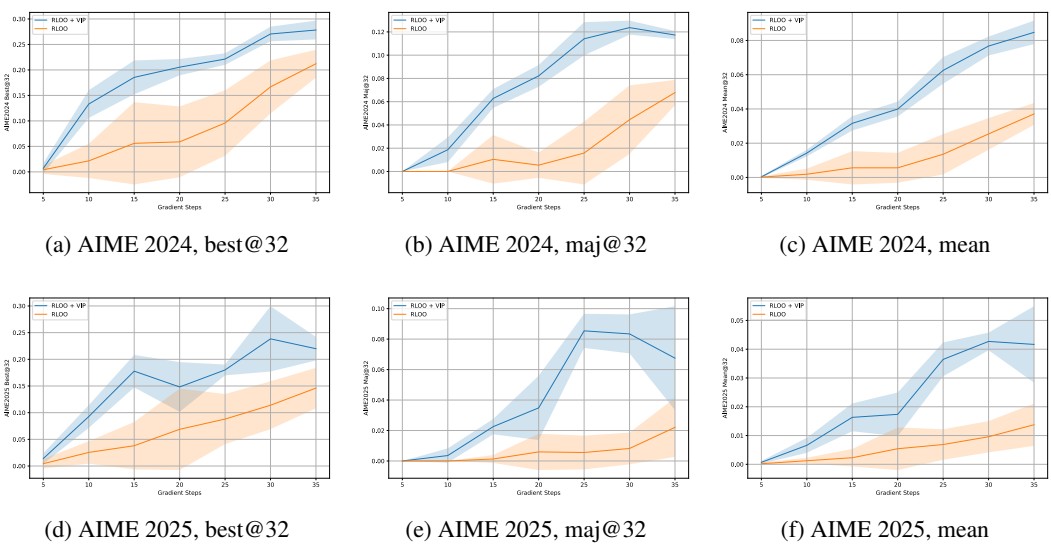

(a) AIME 2024, best@32     (b) AIME 2024, maj@32     (c) AIME 2024, mean

(d) AIME 2025, best@32     (e) AIME 2025, maj@32     (f) AIME 2025, mean

Figure 7: RLOO vs. RLOO+VIP on AIME 2024 and 2025 across different accuracy metrics.