# OpenReview forum: "Adaptive Rollout Allocation for Online Reinforcement Learning with Verifiable Rewards"
_ICLR.cc/2026/Conference — ICLR 2026 Poster_

### Official Review · Reviewer_NY2y · 2025-10-29

**Soundness:** 2
**Presentation:** 2
**Contribution:** 2
**Rating:** 6
**Confidence:** 3

**Summary:**

This paper studies adaptive selection for rollout trajectories for online reinforcement learning (RL)-based language model (LM) training. Specifically, algorithms like GRPO need a batch of rollouts to calculate the (batch normalized) value function, and there are acquisition cost vs. performance tradeoff depending on the number of sample. The paper discuss how to improve the data efficiency of trajectory retrieval by adaptively allocating the rollout budget in a way to minimize the variance of the policy gradient. In particular, the proposed method adapts Baysian approach to estimate the probability of the outcome becomes positive. The experiment results indicates that adding the adaptive rollout components, the algorithm improves the performance compared to uniform rollout allocation.

**Strengths:**

- The motivation for the rollout budget cost and performance tradeoff is well-explained.

- The proposed method can be integrated with off-the-shelf learning algorithms, and the experiment results show an improvement in the result.

- The parameter update for adaptive allocation can be done in an online learning approach using Bayesian updates, and does not seem to require huge computation when new data is added.

**Weaknesses:**

- One concern I have is whether the Bayesian estimation results in biased estimation of the success probability. If I understand correctly, the probability $p$ measures the probability that the outcome sampled on some query results in a positive reward. This probability should depend on the policy's sentence generation probability, and given that the policy is updated during the training phase, the algorithm should be able to handle the non-stationarity of the reward. However, this discussion is not provided, and I wonder whether the estimation is biased towards past policy or the results can change depending on the choice of initial policy.

- Application may be limited to the binary outcome case. It is not obvious how to generalize the idea to the continuous outcome case, especially, how to track the distribution of rewards in such cases.

**Questions:**

- How does the proposed method track the non-stationarity of the expected reward (occurring by the policy's generation probability)?

- How to generalize the idea to the continuous outcome case?

---

> ### Author Response · Authors · 2025-11-21
> **Response to Reviewer NY2y**
>
> We thank the reviewer for taking the time to evaluate our paper and provide positive feedback. We address the reviewer’s remaining questions as follows.
>
> > Whether the Bayesian estimation results in biased estimation of the success probability and non-stationarity of the reward
>
> We appreciate the reviewer’s question regarding whether our Bayesian estimator may be biased toward outdated policies. This concern is directly related to the non-stationarity of the reward distribution, which we explicitly discuss at the beginning of Section 5 (“Predictive Rollout Allocation Strategy”). Handling this non-stationarity is a central design goal of VIP, and the GP-based predictor was chosen precisely for its ability to track distribution shift. Our method avoids bias toward past policies for the following reasons:
>
> - At each iteration, once fresh rollouts from the current policy are collected, the GP posterior is immediately updated using those observations. This ensures that predictions reflect the latest model behavior rather than stale data.
> - Although the initial prior is tied to the initial policy, its effect decays rapidly as more data arrives. During training, predictions are dominated by recent rollouts rather than the initial policy.
> - As discussed in Section 5, success probabilities drift as the policy evolves; this motivates the use of a recursive GP rather than stationary or parametric predictors. The method is built around adapting to this shift.
> - As shown in the new paragraph and Figure 2 (Section 6), we log per-prompt success probabilities over 55 gradient steps of RLOO training and compare GP predictions to Moving Average and Ridge Regression. Across two model scales, the GP consistently achieves the lowest MAE, demonstrating its ability to track evolving success probabilities. Ablation results (Table 3) similarly show degraded performance when replacing the GP with Ridge Regression, confirming that recursive Bayesian updating is essential for handling non-stationarity
>
> > Application may be limited to the binary outcome case. It is not obvious how to generalize the idea to the continuous outcome case
>
> Thank you for raising this important point regarding the generalizability of our approach to continuous rewards. In response, we have added a new appendix section (Appendix E) to the revised manuscript, which formally presents how our method adapts to the continuous reward setting. We summarize the extension here for continuity.
> This extension arises naturally from the analysis in the main paper. Our proofs in Appendix A genuinely present the general formula for the gradient variance for an arbitrary reward distribution. In this setting, the gradient variance for each prompt depends on the variance of its reward distribution, $\mathrm{Var}(\tilde R)$. We then only apply to the binary case, which is popular in RLVR, to derive the propositions in the main paper. Therefore, our variance-aware allocation framework naturally applies to both binary and continuous rewards. The only change required is in the statistic predicted by the GP model: success probability for binary rewards, and reward variance for continuous rewards.
>
> To track the reward variance in the continuous case, we adapt our GP-based prediction model to estimate the per-prompt reward variance using a softplus link function. After observing rollouts, we update the GP with the sample variance and use these predictions in the same budget allocation optimization framework as in the binary case. Formally, given predicted reward variances $\widehat{\mathrm{Var}}(\tilde R_q)$, we define $a_q := \sigma_{Z_q}^2 \widehat{\mathrm{Var}}(\tilde R_q)$ and solve the allocation problems as described in the main text, with only this substitution. The rounding and optimization procedures remain unchanged.
>
> In summary, our approach extends seamlessly to continuous rewards, with the extension following directly from the binary case analysis. For full technical details, please refer to Appendix E in the revised manuscript.

---

### Official Review · Reviewer_6M1P · 2025-10-31

**Soundness:** 2
**Presentation:** 2
**Contribution:** 1
**Rating:** 2
**Confidence:** 4

**Summary:**

The submitted manuscript introduces VIP (Verifiable Importance-based Policy), a method for adaptive rollout allocation in reinforcement learning with verifiable rewards.
Instead of assigning a fixed number of rollouts per prompt, the proposed scheme dynamically allocates rollout budgets based on estimated gradient variance, aiming to improve sample efficiency. This allocation is based on simplified variance computations and the computation of a budget constraint optimization problem. Experiments on math reasoning and question-answering tasks show that VIP achieves better accuracy and reward efficiency compared to uniform rollout allocation.

**Strengths:**

- The paper addresses a relevant and timely problem in reinforcement learning for language models, focusing on efficient rollout allocation under limited budgets.
- The proposed VIP framework is clearly motivated and presented as a practical enhancement to existing RL with verifiable rewards (RLVR) methods.
- The topic is well aligned with current interest in improving training efficiency for large models.
- The proposed idea of allocating samples based on variance estimates, where the unknown dependencies are modeled through a predictive Gaussian process approximation, is both interesting and promising.

**Weaknesses:**

The theoretical analysis is developed under very restrictive assumptions. It sometimes feels as though several of the key challenges of the original setting have been simplified away in order to make the variance computations tractable. While this makes the analysis cleaner, it also raises concerns about the realism and relevance of the resulting conclusions and the potential for counterintuitive sample allocations.

Concern about Assumption 3.1: The assumption $\pi_{old} = \pi_\theta$ effectively removes the off-policy nature of the considered algorithms. Under this setting, the approach seems to reduce to a REINFORCE-style algorithm (with a baseline), where the importance weights are omitted. This simplification has a substantial impact on the resulting variance expressions, since dropping (even clipped) importance weights can alter the underlying variance structure. Consequently, the proposed allocation behaves as if the algorithm were purely on-policy, which does not reflect how these methods are typically used in practice.

There also appears to be a conceptual inconsistency in the motivation. The paper begins by discussing algorithms such as PPO and GRPO, which rely on partially off-policy training through clipped importance weights. Later, however, by adopting Assumption 3.1, the analysis assumes all weights are one and argues that purely on-policy training is preferable. This shift in perspective makes it unclear why the analysis focuses on those off-policy algorithms rather than directly studying the on-policy methods.

Concern about Assumption 4.1: Both parts of this assumption seem rather strong. The independence between $\tilde R$ and $\tilde Z$ is understandable as a simplifying step, but the second part appears difficult to justify in practice. The statistical tests in Appendix B do not provide convincing evidence for these assumptions, they only show that there is insufficient evidence to reject $H_0$. Similar concerns apply to the assumption that all $\tilde Z_q$ share the same variance.

Taken together, Assumptions 3.1 and 4.1 make the variance derivations in Propositions 4.2 and 4.3 relatively straightforward. However, this comes at the cost of removing much of the complexity inherent in the real algorithms. As a result, the computed variances differ substantially from those of the actual gradient estimators used in the experiments.

Finally, while the empirical results show that VIP yields consistent improvements over uniform rollout allocation on several reasoning benchmarks, the gains are difficult to interpret without additional context. In particular, computational costs are not reported (solving the allocation optimization and fitting the Gaussian process introduce extra overhead), and the experiments lack statistical validation such as error bars or multiple runs, making it difficult to assess the robustness of the observed improvements.

**Questions:**

- Under Assumption 3.1, what is the difference between the proposed method and the REINFORCE algorithm?
- In equation (3), where does $\tau$ appear? Shouldn’t $A_j$ depend on $\tau$?
- Why is it feasible to project the vector $H$? While projecting the gradient of $J$ would correspond to a projected gradient method, projecting $H$ itself appears to introduce a bias.
- How do the results in Table 1 change when considering computational cost? Does the budget-allocation procedure introduce significant additional runtime?
- Are the variances of $Z_q$ assumed to be known? If so, why is this assumption considered feasible in practice?

Typos:
- Line 50: …ha[ve] emerged…
- Line 97: Section title should be Related Work[]
- Line 171: However, [o]ur paper…
- Line 266: …denote… denoted…
- In general, I suggest to work over the reference list. In some references there are publisher information missing (Liao et al. 2025, Lin et al. 2025, Sun et al. 2025).
- Some abbreviations are multiple times defined.

---

> ### Author Response · Authors · 2025-11-21
> **Response to Reviewer 6M1P (Part 1)**
>
> We thank the reviewer for the thoughtful comments and the opportunity to clarify our problem setting. Below, we explain why Assumptions 3.1 and 4.1 are appropriate for RLVR and why the resulting variance expressions remain relevant in practice.
>
> > The paper shifts from discussing partly off-policy algorithms to analyzing purely on-policy training, making it unclear why the focus isn’t directly on on-policy methods.
>
> Our goal is to analyze the broad class of RLVR training methods, including RLOO and Dr.GRPO. These algorithms can operate in either an on-policy or a partially off-policy regime, depending on hyperparameters (see [7, Table 1]). For this reason, we did not restrict our discussion to methods designed exclusively for on-policy training.
>
> We acknowledge that the earlier phrasing of Assumption 3.1 may have unintentionally suggested a purely on-policy setting. To prevent confusion, we have refined the assumption to read:
>
> > We assume $r_{j, \tau}(\theta) = 1 (\forall j, \tau)$.
>
> This isolates the importance-ratio simplification without making statements about the underlying policies $\pi_{\text{old}}$ and $\pi$. We also revised the surrounding discussion for clarity.
>
> > The practicality of Assumption 3.1
>
> Assumption 3.1 contains *two independent components*: (a) omitting the KL term, and (b) setting importance ratios to one. We justify both components separately.
>
> **(a) KL removal $(\beta = 0)$:**  In RLHF, KL regularization is essential to prevent reward hacking by learned reward models. RLVR, in contrast, relies on **verifiable rule-based rewards**, whose correctness does not degrade under distribution shift. For this reason, recent RLVR systems routinely remove the KL term and still obtain strong performance (e.g., [4], [5]). Thus, setting $\beta = 0$ is **consistent with current state-of-the-art practice** and not restrictive in this setting.
>
> **(b) Importance ratios:** While existing RLVR work does not explicitly remove importance ratios, in practice the ratios are **empirically extremely close to 1**, making the approximation $r_{j,\tau}(\theta)\approx 1$ both reasonable and practically accurate for variance analysis. This occurs in two training regimes:
>
> * **On-policy behavior is increasingly preferred in RLVR**, both for theoretical stability and to avoid distributional drift ([1], [2], [3]). In strictly on-policy settings, $r_{j,\tau}=1$ holds exactly.
>
> * **Even in partially off-policy RLVR**, importance ratios rarely deviate from 1. This is documented in recent work [6] and is characteristic of LLM post-training where policy updates are incremental.
>
> To validate this phenomenon, we measure importance ratios across 2,560 prompts (8 rollouts each) for Qwen2.5-Math-1.5B. We report the fraction of ratios within $[1 - \alpha, 1 + \alpha]$:
>
> The results are summarized below:
>
> | $\alpha$| Fraction in $[1-\alpha\; 1+\alpha]$ |
> |--------|-----------------------------------------|
> | $5 \times 10^{-2}$ | 97.85% |
> | $5 \times 10^{-3}$ | 82.46% |
> | $5 \times 10^{-4}$ | 71.51% |
>
> These results confirm that **nearly all ratios are extremely close to 1**, supporting the validity of our approximation.
> Finally, **our experiments employ partially off-policy rollouts**, yet our allocation method consistently outperforms baselines. This demonstrates that the insights derived under Assumption 3.1 remain accurate and useful in realistic training settings.
>
>
> > Clarifying Assumption 4.1
>
> We appreciate the reviewer’s attention to the strength of Assumption 4.1 and address the concerns as follows:
>
> * Using statistical tests to validate modeling assumptions is standard practice in empirical ML and statistics. It provides a quantitative assessment of plausibility in real data regimes.
>
> * While failing to reject a null hypothesis does not prove it, our tests are conducted with a *large sample size* (600 prompts × 16 rollouts) and a *strict significance threshold* (0.05), making them sensitive and reliable.
>
> * To further strengthen our justification, we add one additional different test for both correlation assumption in Appendix B.2 and the equal-variance assumption in Appendix B.4. This additional evidence additionally supports the plausibility of our assumptions in our setting.
>
> Together, these results provide robust evidence supporting the plausibility of Assumption 4.1 in our setting.

---

> ### Author Response · Authors · 2025-11-21
> **Response to Reviewer 6M1P (Part 2)**
>
> > Q1: Under Assumption 3.1, what is the difference between the proposed method and REINFORCE?
>
> Under Assumption 3.1, the gradient estimator for group-based RLVR reduces to a REINFORCE-style estimator, but with *group-relative* baselines (mean reward within group), not a global or learned baseline. This structure in Dr. GRPO (GRPO Done Right ) and RLOO (REINFORCE LeaveOne-Out) is empirically more stable and is the standard in RLVR. Our analysis focues on analyzing the gradient variance in this setting.
>
> > Q2: In equation (3), where does $\tau$ appear? Shouldn't $A_j$ depend on $\tau$?
>
> The advantage estimator $A_j$ is token-independent in group-based RLVR, meaning that all tokens in a rollout share the same reward. This design is standard in RLVR methods (e.g., [4], [5], [6]), as noted in our background section. The rationale is inherent to the RLVR setting: rewards are assigned at the sequence level, not at the token level, so the advantage naturally applies uniformly across the entire rollout.
>
> > Q3: Are the variances of Z_q assumed to be known? If so, why is this assumption considered feasible in practice?
>
> In our optimization procedure, we treat the gradient variance as a shared, unknown constant across questions, which is absorbed into the allocation weights. This means our method does not rely on knowing or estimating individual variances for each question.
>
> > Q4: Does the budget-allocation procedure introduce significant additional runtime?
>
> We have added a dedicated paragraph in section 6 and a table to the manuscript (please refer to Table 4), which provides a detailed profile of VIP.
>
> On average, VIP introduces only **1.12%** (1.5B) and **0.83%** (7B) overhead when including cached preprocessing, and only **0.79%** (1.5B) and **0.58%** (7B) during actual training. Because rollout generation dominates overall cost, this relative overhead becomes even smaller for larger models.
>
> > Q5: the experiments lack statistical validation such as error bars or multiple runs, making it difficult to assess the robustness of the observed improvements.
>
> We agree with the reviewer that reporting multiple runs is valuable for assessing robustness. Unfortunately, full retraining with additional seeds is computationally expensive given our current hardware constraints. We also note that prior RLVR and RLHF works typically report single-seed results due to similar computational costs.
> Nevertheless, we acknowledge the importance of this evaluation and are actively running additional seeds. We will include multi-seed results, along with corresponding error bars, in the revised manuscript before the rebuttal deadline.
>
> > About typos:
>
> We thank the reviewer for pointing out these issues. All noted errors have been corrected. Regarding line 50, the original wording is correct because the subject is “a family”, so the verb should be “has emerged”.

---

> ### Author Response · Authors · 2025-11-25
> **Response to Reviewer 6M1P (Part 3)**
>
> > Q6: Why is it feasible to project the vector H? While projecting the gradient of J would correspond to a projected gradient method, projecting H itself appears to introduce a bias.
>
> Our analysis **does not replace the true gradient with the projected one**, nor do we perform a projected-gradient update. The projection is used only as a mathematical device to **reduce a high-dimensional random vector to a scalar** so that we can derive closed-form variance expressions in Propositions 1–2.
> From Eq. 4, we can write the variance of the per-prompt gradient as
>  $$
>  \mathrm{Var}(\tilde G) = \mathrm{Var} \left(\tfrac{1}{n}\sum_j A_j(\\{\tilde o_k\\}) H(\tilde o_j)\right),
>  $$
>  where $H(\tilde o_j)\in\mathbb{R}^d$ can have billions of coordinates. Directly analyzing the full covariance structure is intractable. Hence, we compress $H(\tilde o_j)$ into a scalar
>  $\tilde Z_j$. Any fixed scalarization is acceptable for this purpose:
> projecting onto the all-ones vector, $\tilde Z_j = \mathbf{1}^\top H(\tilde o_j)$, or
> taking the norm, $\tilde Z_j = |H(\tilde o_j)|_2$.
> Both are deterministic functions of $H(\tilde o_j)$. Since the optimization in Eq. 5 only involves factors of the form $p_q(1-p_q)\sigma_Z^2$, and we **assume a constant $\sigma_Z^2$ across prompts**, any constant rescaling of $\tilde Z_j$ simply rescales $\sigma_Z^2$ and **cancels out when comparing prompts**. Thus, **the projection does not introduce bias into the learning algorithm**: the actual training update always uses the full, unprojected gradient, while the projection is used solely for variance analysis.
>
> Regarding the assumption that $\mathrm{Var}(\tilde Z_q)$ is equal across prompts $q$, we emphasize that this is **not** treated as a heuristic. Our **original manuscript already included an empirical homogeneity-of-variance test**. In the revised manuscript, we further strengthen this verification by adding **additional statistical tests** and by evaluating **both scalarizations** of the projected gradient. Specifically, Appendix B3 and B4 now report results from **Levene’s test** and **O’Brien’s test**, conducted on a benchmark of $Q = 600$ questions, each with (n = 16) independent rollouts, using \texttt{Qwen2.5-Math-1.5B}. We evaluate the policy at three training checkpoints (0.0, 0.5, and 1.0 epochs) and perform both tests for $\tilde Z_j = \mathbf{1}^\top H(\tilde o_j)$ and the newly added norm-based scalarization $\tilde Z_j = |H(\tilde o_j)|_2$. As shown in Tables 7 and 8, all global (p)-values from Levene’s test lie between 0.27 and 0.50, and all $p$-values from O’Brien’s test lie between 0.12 and 0.30, consistently exceeding the significance threshold $\alpha=0.05$. Thus, in all settings we **fail to reject the null hypothesis of equal variances across prompts**. These results provide empirical justification for our equal-variance assumption.
>
> **References**
>
> [1] Song, Yuda, et al. "The importance of online data: Understanding preference fine-tuning via coverage." Advances in Neural Information Processing Systems 37 (2024): 12243-12270.
>
> [2] Tang, Yunhao, et al. "RL-finetuning LLMs from on-and off-policy data with a single algorithm." arXiv preprint arXiv:2503.19612 (2025).
>
> [3] Zhang, Wenhao, et al. "On-policy rl meets off-policy experts: Harmonizing supervised fine-tuning and reinforcement learning via dynamic weighting." arXiv preprint arXiv:2508.11408 (2025).
>
> [4] Yu, Qiying, et al. "Dapo: An open-source LLM reinforcement learning system at scale." arXiv preprint arXiv:2503.14476 (2025).
>
> [5] Liu, Zichen, et al. "Understanding r1-zero-like training: A critical perspective." arXiv preprint arXiv:2503.20783 (2025).
>
> [6] Arash Ahmadian, Chris Cremer, Matthias Gallé, Marzieh Fadaee, Julia Kreutzer, Olivier Pietquin, Ahmet Üstün, and Sara Hooker. 2024. Back to Basics: Revisiting REINFORCE-Style Optimization for Learning from Human Feedback in LLMs. In Proceedings of the 62nd Annual Meeting of the Association for Computational Linguistics (Volume 1: Long Papers), pages 12248–12267, Bangkok, Thailand. Association for Computational Linguistics.
>
> [7] Mroueh, Youssef, et al. "Revisiting Group Relative Policy Optimization: Insights into On-Policy and Off-Policy Training." arXiv preprint arXiv:2505.22257 (2025).

---

> ### Author Response · Authors · 2025-11-26
> **Response to Reviewer 6M1P (Part 4)**
>
> We want to report some new results that directly address your last concern in the weakness's section:
>
> > Q6: the experiments lack statistical validation such as error bars or multiple runs, making it difficult to assess the robustness of the observed improvements.
>
> In the revised manuscript, we have added a new section titled **“Training Evolution Comparison”** (Appendix Section G), where we provide statistical validation of robustness across **five independent random seeds** for all four methods (GRPO, RLOO, GRPO+VIP, RLOO+VIP).
>
> Specifically, Figures 6 and 7 report training trajectories under the following controlled setting:
>
> * we retrain $\texttt{Qwen2.5-Math-1.5B}$ using all methods under **five random seeds**,
>
> * using **identical datasets, batch sizes, mini-batch sizes, and a fixed training batch order** to ensure strict comparability across gradient steps;
>
> * We evaluate models **every 5 gradient steps** and report both the **mean and standard deviation** across the five runs.
>
> Across all metrics (best@32, maj@32, mean@32) and two benchmarks (AIME2024 and AIME2025), VIP yields **consistent
> improvements at every step during training**. Additionally, VIP-augmented methods exhibit **lower variance across seeds**, reflecting **greater training stability**.
>
> Due to computational and time constraints, it is not feasible to repeat all experiments in Section 6 with five seeds at the moment. Nevertheless, we believe the newly added results provide strong evidence of the robustness of our method.

---

> ### Author Response · Authors · 2025-11-26
> **Response to Reviewer 6M1P (Summary of Revisions)**
>
> **Dear Reviewer 6M1P,**
>
> Thank you for your thoughtful comments. We have revised the manuscript accordingly, with all updates highlighted in blue. To briefly recap:
>
> * **Assumptions 3.1 and 4.1:** We clarified the writing and added new empirical evidence supporting these assumptions and our variance analysis (revised Assumption 3.1; Appendix B.2 and B.4; Appdendix F).
> * **Runtime profiling:** We added detailed measurements showing that VIP introduces only **0.58%** overhead over the full training run (Section 6, Table 4).
> * **Statistical validation:** We include results across **five random seeds** and **across gradient steps**, as requested (Appendix G).
>
> We believe these updates fully address your concerns. If any additional questions arise, we are happy to clarify. If you are satisfied with the revisions, we kindly ask you to reconsider the rating.
>
> Thank you again for your feedback.
>
> Best regards, Authors

---

> ### Comment · Reviewer_6M1P · 2025-11-26
>
> Dear Authors,
>
> Thank you very much for your careful and thoughtful response to my review. I appreciate the detailed clarifications and the effort you put into addressing my comments. The revised manuscript has significantly improved, particularly in the Numerical Experiments (Section 6 and Appendix G), where the inclusion of multiple random seeds and runtime tracking enhances the credibility of the empirical results. I also understand the computational and time limitations that make further experiments challenging.
>
> That said, while I acknowledge these improvements, my evaluation from a theoretical perspective remains largely unchanged. As a mathematician, I still find the theoretical contribution limited, as the analysis relies on restrictive assumptions that simplify the problem substantially. For this reason, I have increased my overall score to 4 in recognition of the strengthened empirical section, but I have lowered my confidence to 3, as I do not feel fully comfortable assessing the practical impact of the work.
>
> ### Theoretical contribution
>
> My view is unchanged that, from a mathematical standpoint, the theoretical analysis is based on strong simplifying assumptions that make the variance derivations tractable but also limit the generality and interpretability of the results. Although Assumptions 3.1 and 4.1 are supported by statistical tests and empirical examples, I do not find this evidence convincing. Rejecting or failing to reject the null hypothesis in a few examples does not justify treating these assumptions as generally valid, also not in the specific context of LM post-training scenarios.
>
> In particular, I continue to find Assumption 3.1 problematic. I disagree with the statement in line 159 of the revised manuscript that it is ‚not restrictive‘. Even if the empirical ratios in your example are close to one, they are not exactly one, and this relationship could differ substantially in other settings. A possible way forward would be to relax Assumption 3.1 by considering cases where the importance weights are approximately one, and then analyzing how deviations scale with a small parameter $\alpha$ in the notation of Appendix F. This could make the theory more realistic and broadly applicable.
>
> Best regards, Reviewer 6M1P

---

> ### Author Response · Authors · 2025-12-04
> **Final response to R.6M1P**
>
> Dear Reviewer 6M1P,
>
> We are greatly encouraged to see the **rating increase from 2 to 4**. We address the remaining concerns below.
>
> > Although Assumptions 3.1 and 4.1 are supported by statistical tests and empirical examples, I do not find this evidence convincing. Rejecting or failing to reject the null hypothesis in a few examples does not justify treating these assumptions as generally valid, also not in the specific context of LM post-training scenarios.
>
> We agree with the reviewer that no finite set of hypothesis tests can establish the universal validity of an assumption. Our intention is not to claim that Assumption 3.1 holds broadly across all LM post-training settings. Rather, our goal is more modest: to assess whether this assumption is empirically consistent with the RLVR regime studied in this paper. Using statistical tests in this way, as empirical diagnostic tools rather than proofs, is common practice in machine learning and statistics (e.g., [1,2,3]). In this spirit, our empirical analyses in Appendix B suggest that, for RLVR, **Assumption 4.1 is not materially violated in the settings we consider**.
>
> > A possible way forward would be to relax Assumption 3.1 by considering cases where the importance weights are approximately one
>
> We removed the assumption that **$\pi_{old} = \pi_\theta$** and revised Sections 3 and 4 to support a more general setting that includes both on-policy and off-policy updates. In GRPO and RLOO, each update indeed mixes an on-policy component (rollouts from the current batch) and an off-policy component (rollouts from previous batches). However, our research question concerns **how many rollouts to allocate to the current batch given an allocation budget**. The off-policy contributions are fixed once past data are collected and therefore **do not change with the rollout allocation we choose**. Thus, to analyze how allocation affects gradient variance at the current update step, it is both natural and sufficient to isolate the variance arising from the **on-policy** rollouts. The rest of the variance analysis remains unchanged.
>
> We appreciate the reviewer for thoughtful engagement during our rebuttal process and the opportunity to strengthen the rigor of our paper.
>
> Best,
>
> The Authors
>
> Reference:
> 1. Jiao, Zhezhe, and Martin Keller-Ressel. "Emergence of heavy tails in homogenized stochastic gradient descent." Advances in Neural Information Processing Systems 37 (2024): 14066-14092.
> 2. Wang, Jitao, Chengchun Shi, and Zhenke Wu. "A robust test for the stationarity assumption in sequential decision making." International Conference on Machine Learning. PMLR, 2023.
> 3. Sportisse, Aude, et al. "Are labels informative in semi-supervised learning? Estimating and leveraging the missing-data mechanism." International Conference on Machine Learning. PMLR, 2023.

---

### Official Review · Reviewer_dXiM · 2025-11-01

**Soundness:** 3
**Presentation:** 3
**Contribution:** 3
**Rating:** 6
**Confidence:** 2

**Summary:**

The paper proposes adaptive rollout allocation for group-based RL with verifiable rewards: instead of using a fixed number of rollouts per prompt, it predicts each prompt’s success probability and allocates the rollout budget to minimize expected gradient variance. Concretely, the method (“VIP”) fits a lightweight Gaussian Process over prompt embeddings to estimate per-prompt success probabilities, translates them into variance proxies, and solves a convex budget-allocation problem (with an integer rounding heuristic) under a hard rollout budget. Experiments on math reasoning and tool-augmented reasoning show higher accuracy than uniform or heuristic allocation at the same total rollout budget.

**Strengths:**

he paper derives per-prompt gradient-variance formulas for Dr.GRPO and RLOO, motivating variance-aware allocation rather than uniform rollouts.

On AIME24/25 and tool-augmented retrieval, VIP consistently improves accuracy/quality over uniform allocation while keeping the same total number of rollouts.

Replacing either the GP predictor or the optimizer degrades performance, suggesting both components contribute meaningfully.

**Weaknesses:**

Results are reported under the same rollout budget; however, adaptive per-prompt rollout counts can change the number of optimizer steps and total time. The paper should add (a) wall-clock time vs. accuracy and (b) equal-step comparisons to isolate compute-efficiency.

Training a GP and solving the allocation each iteration adds overhead; the paper does not provide detailed runtime/memory profiling vs. uniform baselines.

**Questions:**

See weaknesses.

---

> ### Author Response · Authors · 2025-11-21
> **Response to Reviewer dXiM**
>
> We thank the reviewer for taking the time to evaluate our paper and provide positive feedback. We address the reviewer’s remaining questions as follows.
>
> > Results are reported under the same rollout budget; however, adaptive per-prompt rollout counts can change the number of optimizer steps and total time.
>
> Our experiments already use an equal-compute setting: in every iteration, we enforce $\sum_{q\in\mathcal{B}_t} n_q = C$ (please refer to equations 5, 6, 8,...), where $C$ matches the rollout budget of RLOO and Dr.GRPO (batch size × rollouts per prompt). Thus, all methods see the same total number of rollouts and gradient updates, differing only in how rollouts are distributed across prompts.
>
> > The paper should add wall-clock time vs. accuracy
>
> We believe that wall-clock time is not a reliable criterion for comparing learning algorithms, as it is dominated by system-level factors [1, section 1.2] and by variability in rollout length and tool calls, rather than by the learning algorithm itself. Adaptive allocation mainly reorders when rollouts are consumed, affecting pipeline timing but not the total compute budget. For these reasons, fixed-budget equal-step comparisons are the standard and more interpretable notion of computational efficiency, which we have reported extensively in Table 1 and Table 2.
>
> > Training a GP and solving the allocation each iteration adds overhead; the paper does not provide detailed runtime/memory profiling vs. uniform baselines.
>
> We have added a dedicated paragraph in section 6 and Table 4 to the manuscript, which provides a detailed runtime of VIP.
> On average, VIP introduces only **1.12%** (1.5B) and **0.83%** (7B) overhead when including cached preprocessing, and only **0.79%** (1.5B) and **0.58%** (7B) during actual training. Because rollout generation dominates overall cost, this relative overhead becomes even smaller for larger models.
> Importantly, **VIP requires no additional GPU memory** beyond the uniform baseline. All GP updates and allocation steps run on CPU, and no extra GPU-resident tensors are created. Prompt embeddings and pairwise distances are computed **once before training** and are not part of the training-time memory footprint.
>
> We believe these additions directly address the reviewer’s request for detailed runtime and memory profiling.
>
> References:
> - [1] Cormen, Thomas H., et al. Introduction to algorithms. MIT press, 2022.

---

### Author Response · Authors · 2025-12-04
**Appreciation and Rebuttal Summary**

Dear Reviewers and Area Chair,

We sincerely appreciate the time and effort you dedicated to evaluating our paper and providing valuable feedback. We are grateful that the reviewers recognized several strengths of our paper:
* Our work is a **well-motivated and practical enhancement** to existing RLVR algorithms (R.6M1P, R.NY2y).
* Our proposed framework is **both interesting and promising** (R.6M1P), and **computationally efficient** (R.NY2y).
* Our **experiments and ablation studies strongly support our claims and framework** (R.dXiM, R.6M1P).

We also appreciate **Reviewer 6M1P** for engaging with our rebuttal and for **raising their overall rating from 2 to 4**.

We have incorporated all relevant feedback into the revised version of the paper. All updates are highlighted in **blue** in the main paper and the Appendix. Below, we summarize the changes made during the rebuttal:

> **R.dXiM, R.6M1P**: Does the budget-allocation procedure introduce significant additional runtime?
* We added detailed runtime profiling of VIP and showed that it adds very little overhead to the RL training framework. (Section 6, Table 4)

> **R.NY2y**: Whether the Bayesian estimation results in biased estimation of the success probability and non-stationarity of the reward
* We reported the mean absolute error of our Gaussian Process predictor compared to baselines on a time-series dataset generated during training. The results show that the GP is the most appropriate choice in our setting.  (Section 6, Figure 2).

> **R.6M1P: impractical assumptions.**
* **For Assumption 3.1**: We removed the on-policy assumption and revised Sections 3 and 4 to support a more general setting that includes both on-policy and off-policy updates. We found that our rollout allocation remains unaffected by off-policy updates. (revised assumption 3.1, Sections 3 and 4).

* **For Assumption 4.1**:  We clarified and provided additional empirical justification supporting the practicality of our assumptions (Appendix B2, B4).

> **R.6M1P and R.dXiM**: “the experiments lack statistical validation such as error bars or multiple runs” and “equal-step comparisons to isolate compute-efficiency.”
* We added training-evolution comparisons between RLOO vs. RLOO+VIP and GRPO vs. GRPO+VIP across 5 random seeds. Our algorithms show great improvement across all gradient steps. (Appendix F).

> **R.NY2y**: Application may be limited to the binary outcome case. It is not obvious how to generalize the idea to the continuous outcome case
* We added a detailed framework for extending VIP from verifiable binary rewards to continuous-reward settings. (Appendix E).

Best,

The authors

---

### Meta-Review · Area_Chair_K2A9 · 2026-01-07

**Summary:**

This paper  proposes Variance-Informed Predictive allocation (VIP) as a framework to enhance sampling efficiency in Reinforcement Learning with Verifiable Rewards (RLVR) for LLMs. VIP replaces uniform rollout allocation with a dynamic distribution driven by a lightweight Gaussian Process to estimate per-prompt success probabilities based on recent training data. The probability predictions from the GP are converted into gradient variance estimates used to solve a convex optimization problem to determine an "optimal" allocation of a fixed compute budget across a batch of prompts. Performance on mathematical reasoning, question-answering, and tool-augmented retrieval benchmarks show that VIP consistently achieves higher performance efficiency than standard uniform or heuristic allocation strategies while satisfying the same computational constraints.

The main points raised by the reviewers that contribute to the Accept recommendation are:

Based on the reviews provided in the sources, the main strengths of the paper include:

+ **Relevance:** Reviewers note that the work addresses a relevant problem in contemporary reinforcement learning for LLMs and offers a well-motivated and practical enhancement to existing RLVR methods that can be integrated into off-the-shelf algorithms (6M1P, NY2y).
+ **Empirical results:** VIP consistently improves sampling efficiency and achieves higher accuracy than heuristic allocation strategies across several benchmarks (all reviewers).
+ **Soundness**: VIP is based on solid theoretical foundations (although with some restrictive assumptions) which allow derivations for per-prompt gradient variance. Ablations convincingly show that both the Gaussian Process predictor and optimizer contribute to performance (dXiM, 6M1P).
+ **Novelty**: Using a predictive Gaussian Process to estimate variance for dynamic budget allocation was found an interesting and novel (6M1P).
+ **Efficiency**: VIP's parameter updates utilize a computationally efficient Bayesian approach and requires no significant additional resources (NY2y).

One reviewer (6M1P) raised a number of theoretical issues with the strong assumptions made for the theoretical contributions of the work. During the discussion period this reviewer engaged with the authors and was convinced to raise their score, leaving the evaluation of the empirical contribution of the work to the other reviewers and AC. The empirical results of the work are indeed very interesting, and the paper contains a good mixture of theoretical foundation and empirical evidence which contributes something to the discussion on sample efficiency in RLVR. The recommendation is thus to Accept.

**Reviewer Concerns:**

Two reviewers (dXiM and NY2y) limited their comments mostly to implementation details and questions related to the efficiency of the proposed approach (e.g. the added complexity of training the Gaussian Process and solving the optimization problem at each iteration). These points were adequately addressed in rebuttal by the authors, who provided new analysis of runtime overhead and justified the omission of wall-clock timing measurements based on the fact that the total computational budget is dominated by rollout generation which is constant for all compared methods.

Reviewer 6M1P questioned a number of the assumptions made to allow the theoretical derivations to work, including the stationary policy assumption (which risks violating off-policyness), and the independence assumptions between rollout reward and gradient projections and how the provided statistical tests are inadequate to justify their reasonableness. The authors convincingly defended their assumptions in rebuttal (the reviewer stated they were convinced) and added new details and statistical tests to the manuscript.

Finally, Reviewer NY2y commented that VIP is limited to the binary case and that it is unclear how to extend it to the case of continuous reward. In rebuttal the authors added the theoretical extension to the continuous case.

There were no outstanding issues left unaddressed by the author rebuttal.

**Reviewer Scores:**

+ **R1 (dXiM)**: Raised concerns mostly related to computational efficiency. They would likely have been convinced by the thorough computational efficiency analysis provided in rebuttal and the related revisions to the manuscript.
+ **R2 (6M1P)**: Raised a range of theoretical concerns which were addressed and discussed during the discussion period. Reviewer was convinced to raise their score on the basis of these interactions.
+ **R3 (NY2y)**: Raised questions related to potential bias due to the Bayesian approach used for variance prediction and the generalization of VIP to the continuous, non-binary case. While it is difficult to determine if they would have improved their opinion of the work, their initial opinion was already tending towards Accept.

---

### Decision · Program_Chairs · 2026-01-26

Accept (Poster)